# Analysis of Plasma Emission Experiments and 'Dips'

Spiros Alexiou

Hellenic Army Academy, Varis-Koropiou Avenue, 16673 Vari, Greece; moka1@otenet.gr

**Abstract:** It has been claimed that recent experiments using high-powered lasers have identified dip structures in spectral line profiles in plasmas and that these were successfully used to reliably infer both plasma parameters and information on high and low-frequency turbulence. The analysis of those experiments relies on a flawed theory. In the present work, we computed the line spectra correctly using the parameters inferred in the original papers. The results bear little resemblance to the experimental profiles. The only way to reconcile the parameters deduced in these experiments is to invoke very broadband turbulence, with the relevant distribution functions that are neither known nor measured playing critical roles. Furthermore, the dip positions are shown to be sensitive to details such as field directionality and variations in the frequency and field amplitude. Hence, dips cannot be used to reliably diagnose such plasmas.

**Keywords:** stark broadening; external elecric field; satellites; Floquet

## 1. Introduction

The interaction of an atomic system with oscillatory fields has received a lot of attention, as interesting modifications in the spectral lines arise, such as satellites [1,2]. The presence of a random medium, such as a plasma [3–8], further complicates the picture. Spectral line profiles of highly charged ions in dense plasmas are commonly encountered, and in such spectra, turbulent fields may give rise to considerable modification of the spectral distribution within the line. The study of the turbulent field, and in particular, the characterization of the turbulence via emission line shapes, is of great interest to laser–plasma interaction experiments.

Of special interest is the combination of an oscillatory and a static field. It is reasoned that the static field could be due to static ionic fields and/or low-frequency turbulence, and the oscillatory fields could be due to an external laser and/or high frequency (such as Langmuir) turbulence. The purpose of the present work is to review relevant experiments and analysis [9,10] using the Lyman $\beta(Ly - \beta)$ and $\gamma(Ly - \gamma)$ lines of H-like Si in hot, dense plasma experiments. Specifically, the analysis of these experiments claims that the plasma can be reliably diagnosed in terms of observed (although not clearly above the noise level) "dips". In the present work, we computed the profiles-using the parameters "determined" by the aforementioned analysis rigorously and show that the "dips", if at all present, are not robust, but depend critically on a number of parameters not determined in the experiments, such as field directionality. The present work therefore considers the static and oscillatory fields attributed in the original analysis of the experiments to be non-random, in the sense that all emitters see the same fields, in contrast to the fields generated by the motion of the plasma electrons and ions. Similar analyses have been performed for magnetic fields [11,12]. The present work also considers the effects of variations in the parameters of these non-random fields and the sensitivity of the spectrum and its features, such as the dips. It was not the purpose of this work to diagnose the experimental data, especially in view of the discussion in the next section. One also notes that the criteria used to associate low-frequency with static behavior are often incorrect, as has been shown first in Ref. [13] (the comparison should be with the inverse half-width at half-maximum

(HWHM) time scale, rather with field splittings). In the present work, we also estimated an upper bound for the time scale of fluctuation of the "static" component.

## 2. Review of the Experiments

Recent experiments [9,10] using high-powered lasers claim to show dip structures in the Lyman $\beta (Ly - \beta)$ and $\gamma (Ly - \gamma)$ lines in plasmas that can be used to reliably infer from these both plasmas parameters and information on high and low-frequency turbulence.

The experiments do measure:

- the laser pulse energy delivered at the target;
- the laser intensity at the surface of target;
- the pulse duration;
- the focal spot;
- the targets.

The papers do not mention:

- when the spectra were recorded (how long after the pulse);
- whether optical thickness was an issue (Ref. [10] mentions that FLYCHECK—not the analysis performed—accounted for opacity, if necessary);
- homogeneity of the target area producing the radiation, although Figure 7 of Ref. [10] shows the electron density varying greatly along the target length.

The papers also do not mention whether the plasma parameters were actually measured (e.g., by some independent method, such as Thomson scattering), and it appears that they were not measured but obtained from FLYCHK [14] predictions, although it was mentioned that the electron density is consistent with collisional-radiative calculations [15] in a similar but different experiment. In particular, a single temperature was quoted, presumably referring to electron, ion and Doppler temperatures in thermal equilibrium. Although use was made of the temperature predicted by FLYCHK, it appears that the electron density quoted (which differs from the FLYCHK prediction) was inferred by the dip positions (if present). No explanation is given as to why the FLYCHK temperature prediction is correct if its density prediction is wrong, as the authors claim, and in fact, the line profiles with the electron densities quoted are much narrower than the experimental ones. In addition [9], it is argued (but not measured) that the actual laser intensity at the plasma could be larger than the intensity at the surface because of self-focusing and Raman or Brillouin backscattering. It is mentioned in [9] that instrumental broadening was taken into account in the calculations, but its magnitude is not specified.

The oscillatory field amplitude was also not measured independently but determined from the width of the "dip". Although [9] the authors mentioned two directions, i.e., those of the oscillatory and static fields, which "are not collinear", no mention of angle of observation-dependent profiles was made, nor of whether the profile is different in different directions. In terms of spectral lines, the H-like Si $Ly - \beta$ line was mostly used, although the papers mention and show $Ly - \gamma$ lines, also exhibiting "dips". However, no mention was made of whether the parameters inferred from the $Ly - \beta$ and $Ly - \gamma$ [10] (for the same shot) are consistent. It is also not clear how many shots there were. The authors mentioned initially shots A–D, but also quoted shots 18, 22, and 24 [9]. Last, the ion acoustic wave frequency was not estimated, although to be viewed as static, one would have to show that its frequency is much smaller than the inverse width (HWHM) of the line in question.

## 3. The Present Calculations

The calculations herein were based on those in Ref. [16], which use the interaction picture to first solve the problem (i.e., find the time evolution operator) of the atomic system in the presence of nonrandom fields (i.e., the atomic Hamiltonian and oscillatory and/or static fields) and then apply the random particle fields. This does not assume one-dimensional static and/or oscillatory fields. The first step determines the new "components", i.e., tran-

sition energies and their intensities, and dressing of the effective random particle fields, for the reasons explained in [16]. It makes use of Floquet's theorem, which shows that the time evolution of an atomic system in a periodic field with angular frequency $\Omega$ is a product of a periodic matrix and the exponential $e^{iBt}$ of a constant matrix $B$, which in turns means that, when we Fourier-analyze the periodic matrix, the new spectrum has peaks at positions $\Omega n + b_k$ with $b_k$ representing differences in the eigenvalues of the B-matrix for the upper and lower levels. $n$ is any integer and k is determined by the dimensions of the Hamiltonian matrices of the upper and lower levels. The result is that the line profile is a product of three factors: the atomic dipole matrix element matrix; a matrix S that depends on the nonrandom fields (e.g., oscillatory and static fields) but is independent of the random plasma particle fields, which determines satellite positions and intensities; and a matrix depending on the random plasma interactions and the nonrandom fields that are responsible for the broadening. The last matrix involves the dressing of *all* random particle fields (e.g., the quasistatic component if there is one such) by the nonrandom interaction. This is important because in the calculations of [9,10], a quasistatic particle field distribution is apparently convolved with a quasistatic turbulent (ion acoustic) distribution; however, *all* random fields, whether static or not, are dressed by the periodic field, and this generally results in a slower decay of the autocorrelation function and hence narrowing. The satellite positions are essentially the Blokhintsev satellites, though they include an—in principle nontrivial—shift brought about by the factors that break degeneracy, i.e., fine structure (accounted for in the calculations) and static non-random fields. Thus, the current approach computes the line profile in the presence of non-random oscillatory and static fields (so that the net field is still periodic and Floquet's theorem applies). Should these oscillatory and/or static fields also be random, then an integration with the proper distributions would be warranted. These were not determined in the experiments, and as a result, in the present paper, we investigate the effects of such randomness in the static fields, attributed to low-frequency turbulence. In the current approach, therefore, all physics are included (except for the modification of trajectories and distribution functions of the plasma particles due to the nonrandom fields), and the spectrum is explained in terms of these satellites. Any "dips " are simply the intervals between satellites, to the extent that they have not been filled up by broadening. These "dips" will be further smoothed if the assumed non-random fields are actually random. The calculation method detailed in [16] is still applicable, although the solution for the evolution of the atomic system in the oscillatory plus static field Hamiltonian must be done, in general, for each configuration (i.e., the time history for each emitter contributing to the final profile).

### 3.1. Calculation Details

The calculations were performed using the Floquet eigendecomposition, as discussed in [16]. The turbulent fields (oscillatory and non-oscillatory) are considered, at least initially, as non-stochastic; i.e., all emitters see the same field in time. We discuss and test this assumption later on.

As discussed, we use the interaction picture with the 0th order Hamiltonian, including the atomic Hamiltonian and any nonrandom fields (i.e., the high and low- frequency turbulence). In the presence of an oscillatory non-random field, the line profile in direction **e** is written as:

$$L_{\mathbf{e}}(\omega) = D_{\alpha\beta\beta'\alpha'} \int_{-\infty}^{\infty} dt e^{i\omega t} S_{\alpha_0\alpha_1\beta_1\beta_0\beta\beta'\alpha'\alpha}(t)\{U_{\alpha_1\alpha_0}(t)U^{\dagger}_{\beta_0\beta_1}(t)\} \tag{1}$$

where the dipole term $D$ is purely atomic:

$$D_{\alpha\beta\beta'\alpha'} = \mathbf{d}_{\mathbf{e}\alpha\beta}\cdot\mathbf{d}_{\mathbf{e}\beta'\alpha'} \tag{2}$$

and the plasma-independent (but nonstochastic (e.g., turbulent) field-dependent) matrix $S$ is:

$$S_{\alpha_0\alpha_1\beta_1\beta_0\beta\beta'\alpha'\alpha}(t) = lim_{T\to\infty}\frac{1}{2\pi T}\int_{-T/2}^{T/2}dt_1 U_{\alpha_0\alpha}^{0\dagger}(t_1)U_{\beta\beta_0}^0(t_1)U_{\beta_1\beta'}^{0\dagger}(t+t_1)U_{\alpha'\alpha_1}^0(t+t_1) \quad (3)$$

$S$ determines the satellite structure and intensity, $D$ determines the total line intensity and the plasma-dependent part $\{\dots\}$ determines the broadening of each satellite. In the case without a oscillatory field, $U_{ij}^0(t) = \delta_{ij}exp(-i\omega_i t)$

For a purely periodic field, a key quantity is the product

$$\bar{S}_{\alpha_0\alpha_1\beta_1\beta_0} = D_{\alpha\beta\beta'\alpha'}^{\mathbf{e}}S_{\alpha_0\alpha_1\beta_1\beta_0\beta\beta'\alpha'\alpha}(t) = \sum_k\sum_{n=-\infty}^{\infty}Q_{nk}e^{i(\Omega n + \Delta_k)t} \quad (4)$$

with n running over all integers (corresponding to Blokhintsev satellites) and $k$ running over the distinct combinations $k$ of differences in the upper–lower Floquet exponents. The line profile is thus seen to consist of components at positions $\Omega n + \Delta_k$ and intensities $Q_{nk}$. For non-stochastic turbulence (i.e., all emitters see the same turbulent field), $U_0$, and hence $Q_{nk}$ and $\Delta_k$, are computed once and for all when the atomic evolution in the turbulent field is solved.

This results in a modified satellite structure: If the plasma-dependent quantity $\{U_{\alpha_1\alpha_0}(t)U_{\beta_0\beta_1}^{\dagger}(t)\}$ is Fourier-analyzed in terms of functions $u_j(\omega)$, then the profile qualitatively consists of linear combinations of $u_j(\omega - n\Omega - \Delta_k)$. The width and shift of each such component (combination of n and $k$) are determined purely by the decay of $\{U_{\alpha_1\alpha_0}(t)U_{\beta_0\beta_1}^{\dagger}(t)\}$, where $\{\dots\}$ denotes the quantum statistical average over the plasma, and $U(t)$ is determined as the solution of

$$\frac{dU}{dt} = -\frac{\imath}{\hbar}U_0^{\dagger}(t)V(t)U_0(t), U(0) = I \quad (5)$$

with $V(t)$ being the random interaction between plasma electrons and ions.

Therefore, first, the time evolution operator $U_0$ for the atomic system in question (e.g., the $n = 4$ for $L_\gamma$ and the $n = 3$ level for $L_\beta$) was computed under the influence of a linearly polarized field $E_0cos\omega_p t$ in the z-direction and a static field **F** with $\omega_p$ the electron plasma frequency corresponding to the electron density in question. This has a Floquet structure and enables us to compute the peaks in the spectrum, while keeping only the satellites in the spectral region of the experiments. To compute the autocorrelation function, 2000 configurations were used, each one involving *all* electrons and ions, moving in hyperbolic trajectories [1] that become relevant [17–19] (i.e., come closer than three screening lengths) at any time in (0,T), where T is a time such that either the autocorrelation function $C(T) \ll C(0)$ or an asymptotic tail is recognized (whichever comes first). This means that a number of autocorrelation functions (each one corresponding to a different peak) were computed, Fourier-transformed and finally added to produce the final profile. This was done separately for the $\sigma$ and $\pi$ components (with respect to the oscillatory field). Fine structure was fully included, and a Doppler convolution was performed for each component (i.e., the computed autocorrelation function was multiplied by the Doppler autocorrelaton function before Fourier-transforming). All calculations thus include Doppler broadening, corresponding to the Doppler temperature assumption of the specific calculation (which is the same as the temperature in the "main" calculation).

### 3.2. Outline of the Different Calculations

For each profile presented in Refs. [9,10], a number of different calculations were performed. Here, we briefly outline them and their rationale. First, we performed the standard Stark-broadening calculations (i.e., no oscillatory field and no extra static field),

with (FS) and without (WFS) fine structure. Except for WFS, all other calculations take into account fine structure. In addition, there was a calculation labeled "main", which includes the effect of a linearly polarized oscillatory field at the electron plasma frequency of the electron density of the experiment and with the amplitude stated in Refs. [9,10] and a noncollinear static field with strength as given in Refs. [9,10]. These references make no mention of the strengths of the static fields parallel and perpendicular to the oscillatory field, taken to be along the z-direction. These parallel and perpendicular components are stated in our calculations. Next, the main calculation was repeated with ions alone (i.e., electron broadening is neglected), in order to access the relative importance of electron and ion broadening for the new, renormalized atomic system. For the $Ly - \beta$ calculations, the effect of electrons is shown to be small, as expected, whereas for the $Ly - \gamma$, it is significantly more important, also as expected. Note that in all other (i.e., except the "ions only") calculations, electron and ion broadening were computed simultaneously, although neglecting electrons would speed up the calculation significantly. All calculations used the collision-time statistics method [17] with Seidel's improvement [18,19].

Additional calculations were performed for each case. Specifically:

To test the effects of a different mixtures of parallel and perpendicular static fields, the main calculation (MC) was rerun with the same static field magnitude, but (a) perpendicular to the oscillatory field or (b) parallel to the oscillatory field. This gives an idea of the sensitivity of the profile in the direction of the static field.

It is clear that the experimental profiles cannot be produced by stochastic thermal fields (i.e., due to plasma electrons and ions) plus non-random turbulent fields. Hence, randomness in the turbulent field is investigated as follows: To test the effect of a random static (but not oscillatory) field, the MC was repeated with the perpendicular (to the oscillatory field) component (a) zero and (b) twice the size of the component of the MC. The point is that if the static field is random with some mean values quoted in the experiment, then we would obtain an average of these values, and the differences between MC and (a) and (b) allow a better understanding of what the final profile would look like. In principle, we could average over different turbulent field realizations to obtain a final profile. The main problem with this is that the distribution functions of the turbulent fields are neither known nor measured. Furthermore, the "dips" claimed in Refs. [9,10] are not supposed to depend on these distribution functions.

Since the calculations in Refs. [9,10] appear to add the static field to the particle (assumed) quasistatic component, we also repeated the MC calculation so that $U_0$ wouldd be correctly used in Equations (3) and (4), which give the correct satellite positions and intensities, but not in Equation (5), where $U_0(t)$ was taken to be the unit matrix (no dressing). As expected, this resulted in wider profiles, but this effect is completely inadequate to match the experimental profile (or the calculations of Refs. [9,10]).

Since the ion temperature was not measured, we also repeated calculations with cold ions, i.e., 1 eV ionic temperature. Clearly the lower ionic temperature will only decrease the widths and thus exacerbate the differences with the very wide experimental profiles.

All results had their profiles scaled so that the peak is one, except for the profiles accounting for the field anisotropy, where the $\sigma$ component has a peak of one and the $\pi$ component preserves the $\sigma$ to $\pi$ ratio.

We also plot the autocorrelation functions (the Fourier transforms of their profiles) for various components. We say "various" to mean that all distinct time scales are shown. The components not shown (for clarity of the figure) have autocorrelation functions that either coincide with or are very close to the ones that are shown. These decay with time, and their decay allows the estimation of upper bounds for the "low-frequency" turbulence, which were taken as static in the calculation and in Refs. [9,10]. In addition, we show the autocorrelation functions of the two finely structured components of the lines in question

without the oscillatory or static fields, which show faster decay and hence larger width than the main calculation. This line narrowing by a fast oscillatory field is important and well understood [20–23]. However, dressing is not the only reason for the narrowing of the profiles: This is illustrated also by plotting the autocorrelation functions of the various components from the "no dressing" calculation, i.e., with dressing turned off but the components (i.e., the S-factor) correctly computed, as previously discussed. This decays faster than the main calculation but slower than the calculations without oscillatory and static fields. The point is that the emergence of new components quickly slows the decay rate because probability amplitude (off-diagonal U-matrices) is distributed among more components.

Another qualitatively important aspect that emerges from the plots of the autocorrelation functions is the absence of the characteristic $Ly_\beta$ dip—i.e., a region where the autocorrelation function becomes negative. This is a combination of two effects: First, in the usual case, if the interaction is strong, it causes a large drop in the diagonal upper level U-matrix element, and hence the autocorrelation function on a time scale short enough for ions to be effectively static, and this static solution is responsible for the dip. With the renormalization due to external fields, the decay is significantly slower [20–22], so that the field cannot be viewed as static. The second reason is that the atomic structure is renormalized, and as a result, the selection rules are modified.

## 4. $Ly - \beta$ Calculations

For $L_\beta$ 5, "main" and corresponding auxilliary calculations (labeled A–E) were performed, with the parameters described in Table 1.

**Table 1.** Ly-$\beta$ calculation parameters.

| Calculation | El.Density ($\times 10^{21}$e/cm$^3$) | T (eV) | F (GV/cm) | $E_0$ (GV/cm) | Ref. |
|:---:|:---:|:---:|:---:|:---:|:---:|
| A | 22 | 600 | 4.8 | 0.7 | Figure 5a in [10] |
| B | 22 | 550 | 4.4 | 0.5 | Figure 5b in [10] |
| C | 22 | 600 | 4.9 | 0.6 | Figure 5c in [10] |
| D | 6.6 | 550 | 1.41 | 2 | Figure 5d in [10] |
| E | 17.4 | 500 | 3.9 | 1 | Figure 3b in [10] |

*4.1. Case A*

4.1.1. Main Calculation A

Figure 1 displays the results of the main calculation (MC) for the $\sigma$ (solid line) and $\pi$ (dashed line) profiles. The static field was taken to be 4.12 and 2.46 GV/cm in the x (perpendicular) and z (parallel to the oscillatory field) directions, respectively. Shown as well are the results without static or oscillatory fields (i.e., the pure thermal profiles) with (dotted line) and without (dashed–dotted line) fine structure and the experimental result (dot–double-dashed line) of Refs. [9,10]. The results show that the experimental profile is much larger than the calculated profile, that satellites (and associated dips) are clearly visible and that the dips do not coincide with the positions calculated by the method of Refs. [9,10].

As expected, and as may be seen in Figure 2, the profile is determined by ion broadening; electrons only slightly fill in the dips between the various peaks. Nevertheless, although neglecting electrons would speed up the calculation significantly, all other calculations fully include electron broadening too.

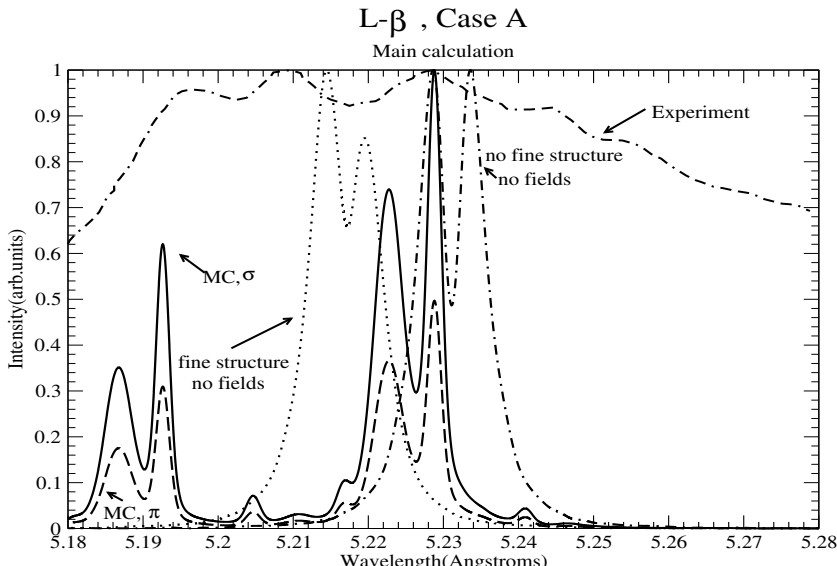

**Figure 1.** Main calculation for $Ly - \beta$, case A. Shown are the $\sigma$ (solid line) and $\pi$ (dashed line) profiles for the main calculation; the profiles for the calculations without oscillatory and static fields, with (dotted line) and without (dashed−dotted line) fine structure; and the experimental profile (dot−double−dashed line) of Ref. [10].

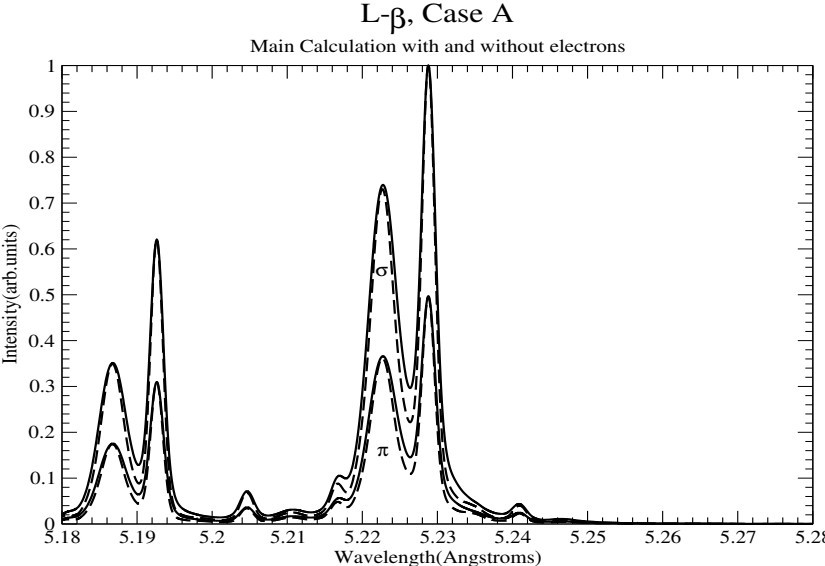

**Figure 2.** Main calculation for $Ly - \beta$, case A, with only ion broadening accounted for (i.e., no electrons). Shown are the main calculation $\sigma$ and $\pi$ profiles (solid line), along with the corresponding profiles for the calculations without electron broadening (dashed line).

### 4.1.2. Effect of Dressing

As discussed above, the effect of dressing is investigated here also in Figure 3: In addition to the main calculation (solid line), a "no dressing" calculation (dashed line) is shown, where the matrix $S$ which described the satellite positions and intensities was correctly computed, but the emitter–plasma interaction in the Schrödinger equation was (incorrectly) not dressed by the nonrandom static and oscillatory fields. This resulted in filling the dips. A well-known fact is that additional broadening mechanisms for lines with no central components tend to fill in the dip.

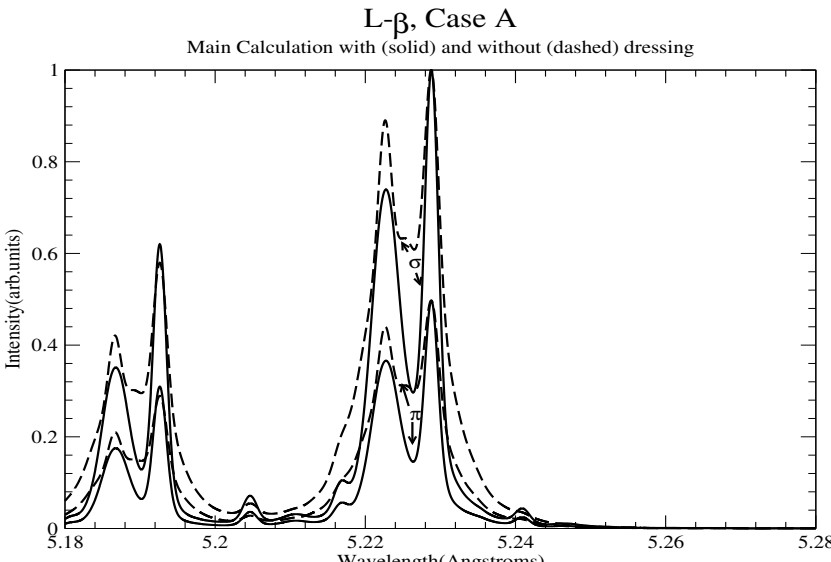

**Figure 3.** Effect of dressing for $Ly - \beta$, case A. Shown are $\sigma$ and $\pi$ profiles for the main calculation (solid line) and the "no dressing" calculation (dashed line); i.e., the matrix $S$ which described the satellite positions and intensities was correctly computed, but the emitter–plasma interaction was (incorrectly) not dressed by the nonrandom static and oscillatory fields.

### 4.1.3. Directional Dependence

Two extra calculations (Figures 4 and 5 for the $\sigma$ and $\pi$ components respectively ) were done: By keeping the electron density and temperature fixed as in the main calculation (solid line); assuming identical electron, ion and Doppler temperatures; and assuming a Langmuir nonrandom field in the z-direction, the profile was calculated with a static nonrandom field of 4.8 GV/cm which (a) is parallel to the Langmuir field (dotted line) and (b) is perpendicular to the Langmuir field (dashed line). (b) is very similar to the MC, due to the rather small z-component. (a), however, is very different. Thus, direction *does* matter, which again speaks against a robust dip diagnostic as in Refs. [9,10]. It is clear that information on field direction can in principle be extracted from the profiles, though the experiments give no such information.

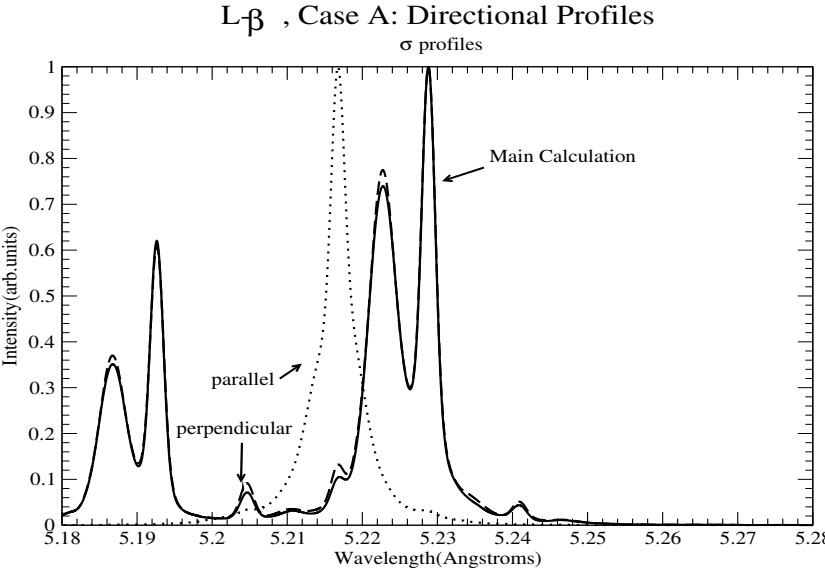

**Figure 4.** Directional effects for $Ly - \beta$, case A. Shown are $\sigma$ profiles for the main calculation (solid line) and the entire static field (4.8 GV/cm) when parallel (dotted line) or perpendicular (dashed line) to the oscillatory field.

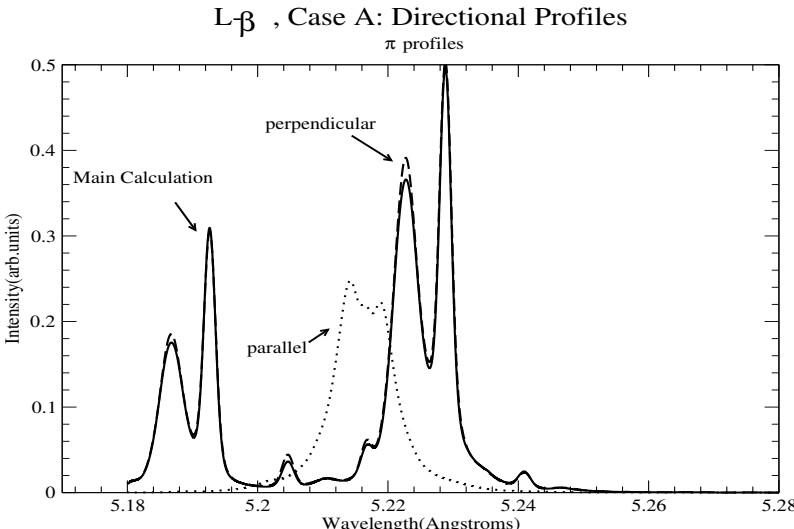

**Figure 5.** Directional effects for $Ly - \beta$, case A. Shown are $\pi$ profiles for the main calculation (solid line) and the entire static field (4.8 GV/cm) when parallel (dotted line) or perpendicular (dashed line) to the oscillatory field.

### 4.1.4. Effect of Randomness of Static Field

To test the effect of randomness in only the static field, the main calculation was repeated, while keeping the static component parallel to the oscillatory field as in the main calculation and varying the perpendicular component. To this effect, two calculations are shown in Figures 6 and 7: In both calculations, the static component parallel to the oscillatory field was 2.46 GV/cm, but in the first, labeled '0 field', the perpendicular component was 0, whereas in the second, labeled 'x2', it was taken to be 8.24 GV/cm, twice the value of the main calculation. The calculations show that the profile—in particular, the satellite intensity—is very sensitive to the static field. Therefore, if the experimental profile is to be attributed to a combination of an oscillatory and a static field, as proposed in Refs. [9,10], then two things are clear: first, a broad distribution of static field intensities would be required and second, any "dips" would critically depend on the details of that distribution and hence by themselves (without the distribution) cannot provide any reliable diagnostics.

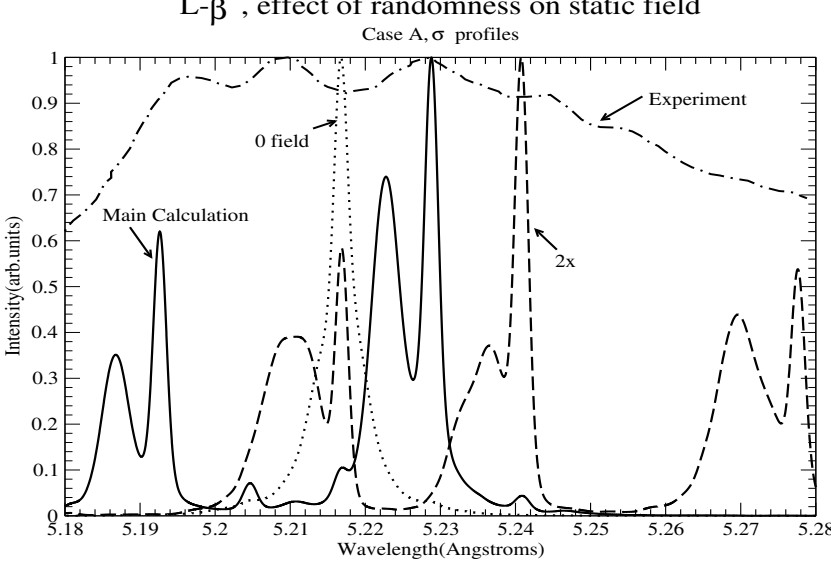

**Figure 6.** Randomness of static component perpendicular to the oscillatory field for $Ly - \beta$, case A. Shown are $\sigma$ profiles for the main calculation (solid line), the component perpendicular to oscillatory

field 0 (0 field-dotted) and the component perpendicular to the oscillatory field but twice the size of that in the main calculation, i.e., 8.24 GV/cm (2x-dashed). The static component perpendicular to the oscillatory field is the same as in the main calculation, 2.46 GV/cm. Shown as well is the experimental profile (dashed–dotted line).

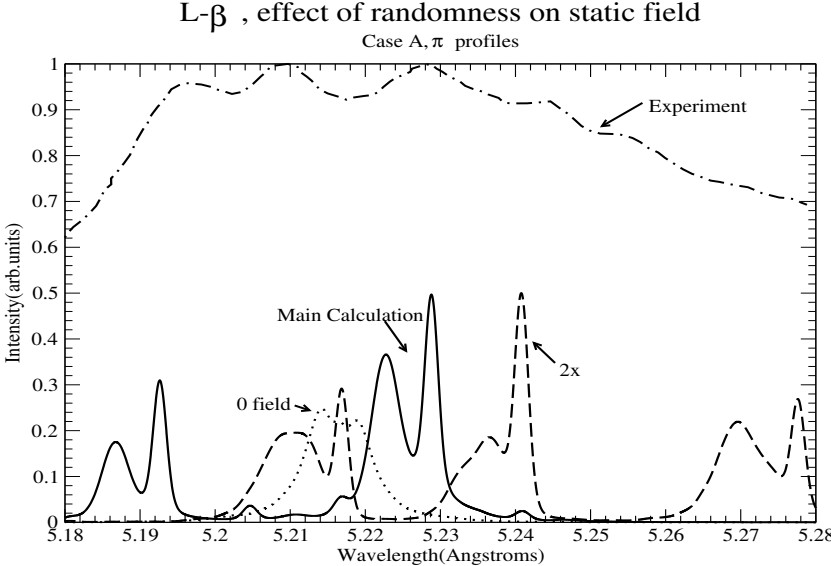

**Figure 7.** Randomness of static component perpendicular to the oscillatory field for $Ly - \beta$, case A. Shown are $\pi$ profiles for the main calculation (solid line), the component perpendicular to oscillatory field 0 (0 field-dotted) and the component perpendicular to the oscillatory field but twice the size of that in the main calculation, i.e., 8.24 GV/cm (2x-dashed). The static component perpendicular to the oscillatory field is the same as in the main calculation, 2.46 GV/cm. Shown as well is the experimental profile (dashed–dotted line).

### 4.1.5. Cold Ions

Figure 8 shows the main calculation repeated with cold (T = 1 eV) ions. All other parameters stayed the same as in the main calculation. Shown are the $\sigma$ (solid line) and $\pi$ (dashed–dotted line) main calculation profiles and the corresponding $\sigma$ (dashed line) and $\pi$ (dotted line) profiles for an ion temperature of 1 eV. As expected, the regions between the peaks are filled less, hence exacerbating the differences from the experiment.

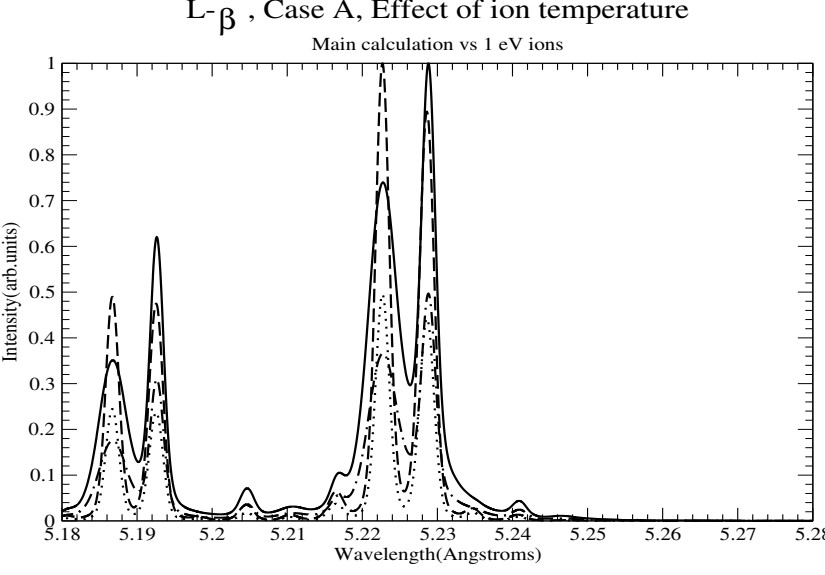

**Figure 8.** $\sigma$ (solid line) and $\pi$ (dashed–dotted line) main calculation profiles, and the corresponding $\sigma$ (dashed line) and $\pi$ (dotted line) profiles for an ion temperature of 1 eV.

### 4.1.6. Static Timescale

Figure 9 shows the autocorrelation functions C(t) of the different components (peaks) for the main calculation. The solid lines correspond to the "main calculation" components; the dotted lines correspond to the two finely structured components in the absence of oscillatory and static fields. The dashed–dotted line corresponds to the calculation without fine structure and without oscillatory or static fields. The dashed lines are the components in the calculation where dressing was turned off. To be static, a field must vary at worst only slowly on the timescale where these C(t) are appreciable; i.e., the variation must be on time scales longer than 3 fs (the slowest decay time scale). We thus infer that turbulence can be considered static if it is associated with frequencies $\ll 0.3 \times 10^{15}$ s$^{-1}$. Note that in contrast to the normal case (i.e., without a periodic field), where C(t) becomes negative, leading to a central dip in the profile, C(t) in the main calculation does not become negative. However, although dressing does make a significant difference, as illustrated by the difference in the decay of the solid and dashed components, it is not solely responsible for the slower decay and smaller widths: The emergence of new channels also effectively delays the decay.

L-β , Case A: Autocorrelation functions
of various components

**Figure 9.** Autocorrelation functions C(t) for the various components (corresponding to different Blokhintsev–Floquet peaks) of the $L_\beta$, case A calculation. The solid lines correspond to the "main calculation" components; the dotted lines correspond to the two finely structured components in the absence of oscillatory and static fields. The dashed–dotted line corresponds to the calculation without fine structure and without oscillatory or static fields. The dashed lines are the components in the calculation where dressing was turned off.

### 4.2. Case B

#### 4.2.1. Main Calculation B

Figure 10 displays the results of the main calculation (MC) for the $\sigma$ (solid line) and $\pi$ (dashed line) profiles. The static field was taken to be 3.22 and 3 GV/cm in the x and z-directions (parallel to the oscillatory field), respectively. Shown as well are the results without static or oscillatory fields (i.e., the pure thermal profiles), with (dotted line) and without (dashed–dotted line) fine structure; and the experimental results (dot–double-dashed line) of Refs. [9,10]. As before, the experimental profile is much wider than for the calculation, and satellites (and associated dips-at different positions though-) are clearly visible, in contradiction to the calculations of Refs. [9,10].

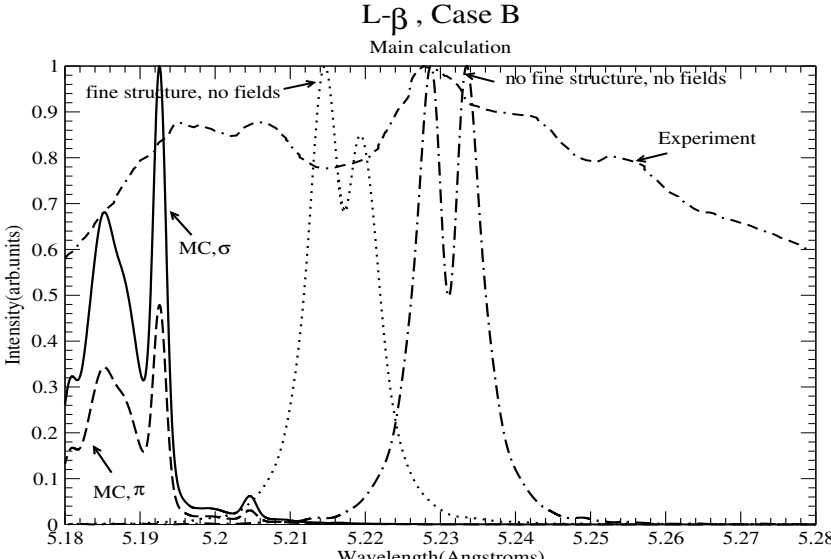

**Figure 10.** Main calculation for $Ly - \beta$, case B. Shown are the $\sigma$ (solid line) and $\pi$ (dashed line) profiles for the main calculation; the profile without oscillatory and static fields, with (dotted line) and without (dashed–dotted line) fine structure; and the experimental profile (dot–double-dashed line) of Refs. [10].

Note that, although the parameters are similar to those in Case A, the spectrum is very different, because of the sensitivity of the 0th order (i.e., without plasma electrons and ions) solution. Figure 11 illustrates by showing the positions and relative intensities of the various components for the two cases, A (+) and B (*).

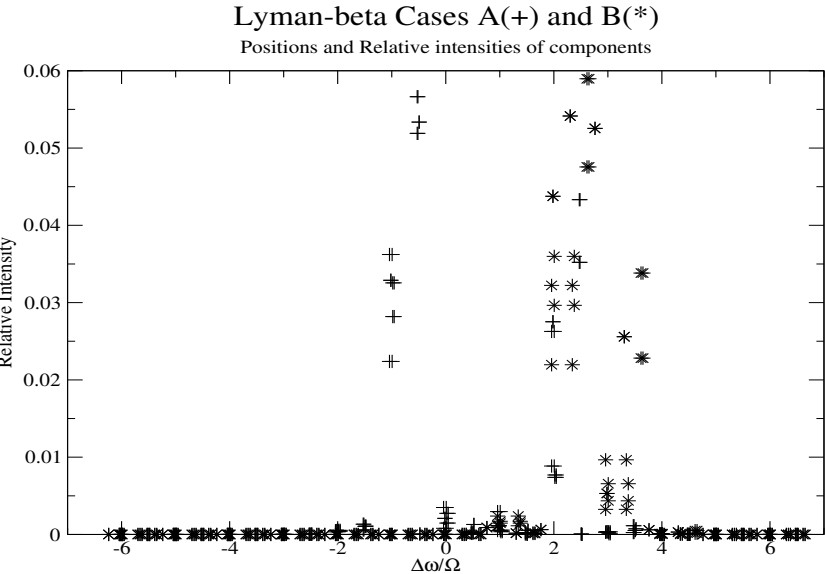

**Figure 11.** Positions and intensities for the main calculation for $Ly - \beta$, cases A(+) and B(*). $\Delta\omega$ is measured from the line center of gravity.

As expected, and as may be seen in Figure 12, the profile is determined by ion broadening, and electrons only slightly fill in the regions between the various peaks. Nevertheless, although neglecting electrons would speed up the calculation significantly, all calculations fully include electron broadening too.

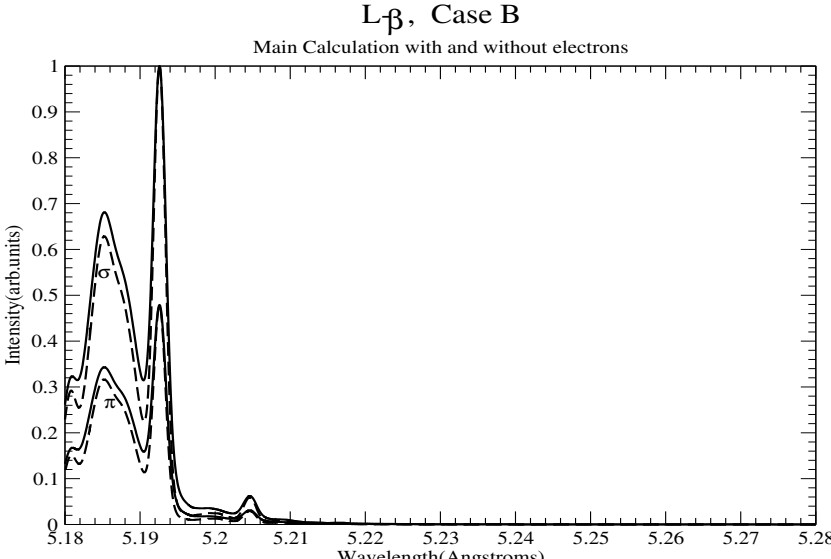

**Figure 12.** Main Calculation for $Ly - \beta$, case B with only ion broadening accounted for (i.e., no electrons). Shown are the main calculation $\sigma$ and $\pi$ profiles (solid line), together with the profiles for the calculations without electron broadening (dashed line).

### 4.2.2. Effect of Dressing

As discussed above, the effect of dressing is investigated here also in Figure 13: In addition to the main calculation (solid line), a "no dressing" calculation (dashed line) is shown, where the matrix $S$, which described the satellite positions and intensities, was correctly computed, but the emitter–plasma interaction in the Schrödinger equation was (incorrectly) not dressed by the nonrandom static and oscillatory fields.

As expected, this resulted in a broader line, but this effect is again completely inadequate to match the profile (or the calculations of Refs. [9,10]).

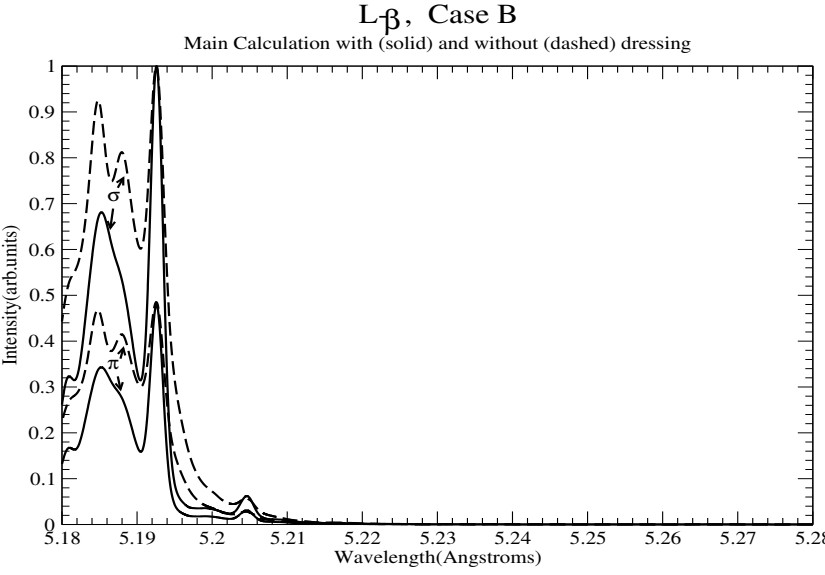

**Figure 13.** Effect of dressing for $Ly - \beta$, case B. Shown are $\sigma$ and $\pi$ profiles for the main calculation (solid line) and the "no dressing" calculation (dashed line), i.e., the matrix $S$, which described the satellite positions and intensities, was correctly computed, but the emitter–plasma interaction was (incorrectly) not dressed by the nonrandom static and oscillatory fields.

### 4.2.3. Directional Dependence

Two extra calculations were performed (Figure 14 for the $\sigma$ and Figure 15 for the $\pi$ components respectively) : By keeping the electron density and temperature fixed, as in the main calculation (solid line); assuming identical electron, ion and Doppler temperatures; and assuming a Langmuir nonrandom field in the z-direction, the profile was calculated with a static nonrandom field of 4.4 GV/cm, which (a) is parallel to the Langmuir field (dotted line) and (b) is perpendicular to the Langmuir field (dashed line). Again, the profiles are quite sensitive to direction.

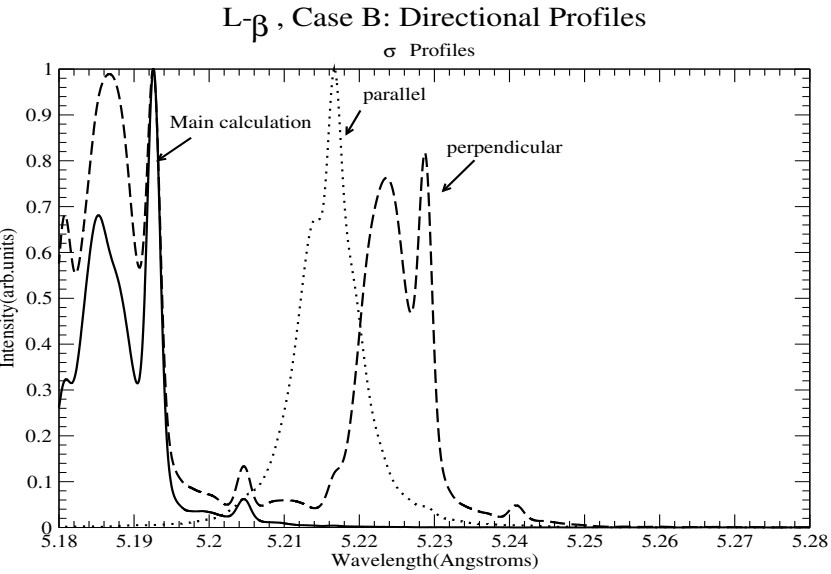

**Figure 14.** Directional effects for $Ly - \beta$, case B. Shown are $\sigma$ profiles for the main calculation (solid line) and the entire static field (4.4 GV/cm) when parallel (dotted line) or perpendicular (dashed line) to the oscillatory field.

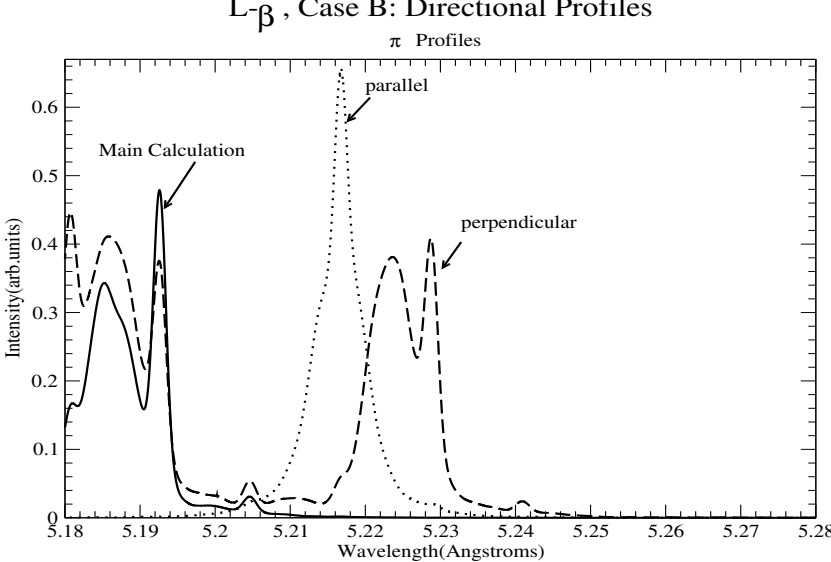

**Figure 15.** Directional Effects for $Ly - \beta$, case B. Shown are $\pi$ profiles for the main calculation (solid line) and the entire static field (4.4 GV/cm) when parallel (dotted line) or perpendicular (dashed line) to the oscillatory field.

### 4.2.4. Effect of Randomness of Static Field

Figures 16 and 17 show the $\sigma$ and $\pi$ profiles, respectively, for the main calculation, and the calculation with the perpendicular static field (a) at zero and (b) at twice its MC value. Again, the dips are observed in very different locations.

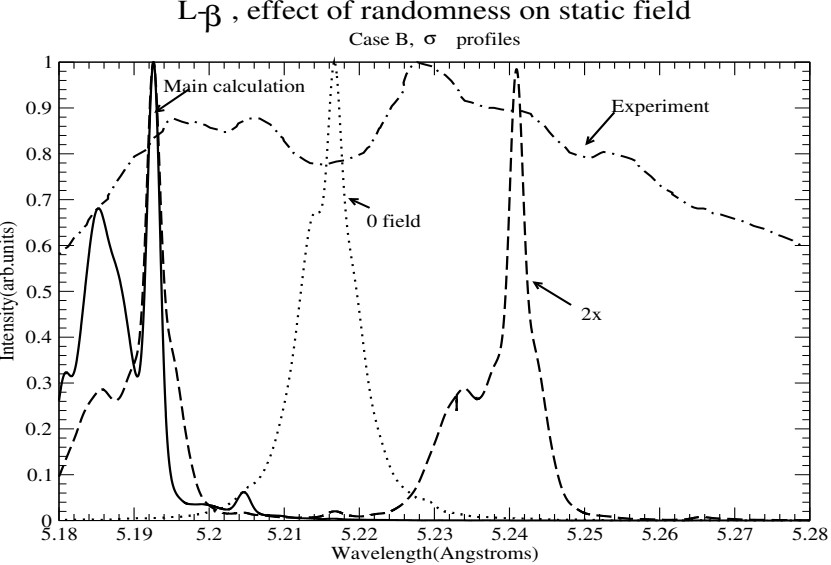

**Figure 16.** Randomness of static component perpendicular to the oscillatory field for $Ly - \beta$, case B. Shown are $\sigma$ profiles for the main calculation (solid line), the component perpendicular to oscillatory field 0 (0 field-dotted) and the component perpendicular to the oscillatory field but twice the size of that in the main calculation, i.e., 6.44 GV/cm (2x-dashed). The static component perpendicular to the oscillatory field is the same as in the main calculation, 3 GV/cm. Shown as well is the experimental profile (dashed–dotted line).

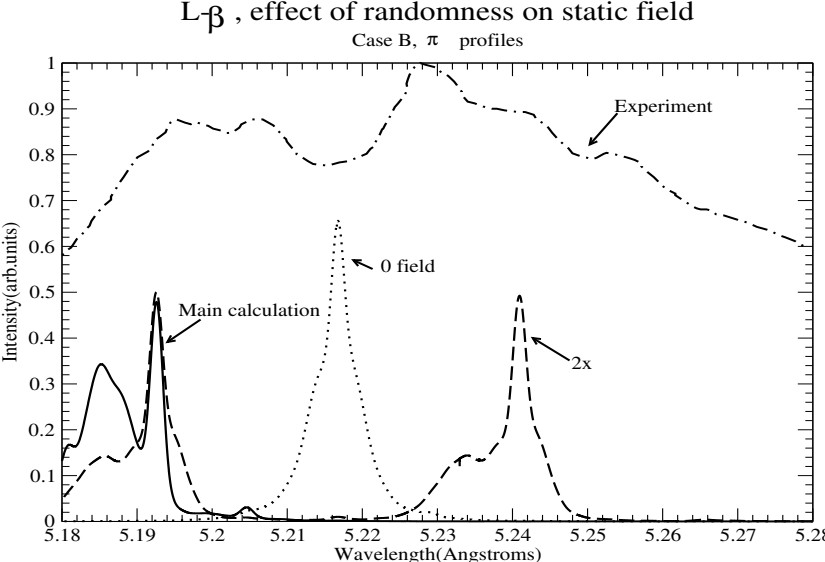

**Figure 17.** Randomness of static component perpendicular to the oscillatory field for $Ly - \beta$, case B. Shown are $\pi$ profiles for the main calculation (solid line), the component perpendicular to oscillatory field 0 (0 field-dotted) and the component perpendicular to the oscillatory field but twice the size of that in the main calculation, i.e., 6.44 GV/cm (2x-dashed). The static component perpendicular to the oscillatory field is the same as in the main calculation, 3 GV/cm. Shown as well is the experimental profile (dashed–dotted line).

#### 4.2.5. Cold Ions

Figure 18 shows the main calculation repeated with cold (T = 1 eV) ions. All other parameters stayed the same as in the main calculation. Shown are the $\sigma$ (solid line) and $\pi$ (dashed–dotted line) main calculation profiles and the corresponding $\sigma$ (dashed line) and $\pi$ (dotted line) profiles for an ion temperature of 1 eV. As expected, the regions between the peaks are filled less, hence exacerbating the differences from the experiment. Specifically, components that merged in MC now appear distinct.

### L-β, Case B, Effect of ion temperature

Main calculation vs 1 ev ions

**Figure 18.** $\sigma$ (solid line) and $\pi$ (dashed–dotted line) main calculation profiles and the corresponding $\sigma$ (dashed line) and $\pi$ (dotted line) profiles for an ion temperature of 1 eV.

#### 4.2.6. Static Timescale

Figure 19 shows the autocorrelation functions C(t) of the different components (peaks) for the main calculation. The solid lines correspond to the "main calculation" components, the dotted lines correspond to the two finely structured components in the absence of oscillatory and static fields. The dashed–dotted line corresponds to the calculation without fine structure and without oscillatory or static fields. The dashed lines are the components in the calculation where dressing was turned off. To be static, a field must vary at worst only slowly on the timescale where these C(t) are appreciable; i.e., the variation must be on time scales longer than 3 fs. We thus infer that turbulence can be considered static if it is associated with frequencies $\ll 0.3 \times 10^{15}$ s$^{-1}$. Again, in contrast to the normal case (i.e., without a periodic field), where C(t) becomes negative, leading to a central dip in the profile, C(t) in the main calculation did not become negative. Again, although dressing does make a significant difference, as illustrated by the difference in the decay of the solid and dashed components, it is not solely responsible for the slower decay and smaller widths: the emergence of new channels also effectively delays the decay.

#### 4.3. Case C

#### 4.3.1. Main Calculation C

For this case, the perpendicular and parallel static field components were taken to be 3.87 and 3 GV/cm, respectively.

Figure 20 displays the results of the main calculation (MC) for the $\sigma$ (solid line) and $\pi$ (dashed line) profiles. Shown as well are the results without static or oscillatory fields (i.e., the pure thermal profiles) with (dotted line) or without (dashed–dotted line) fine structure, and the experimental results (dot–double-dashed line) of Refs. [9,10]. Again, the results show that the experimental profile is much wider than the calculation, and that satellites

(and associated dips) are clearly visible. As expected, and as may be seen in Figure 21, the profile is determined by ion broadening; the electrons only slightly fill in the dips between the various peaks.

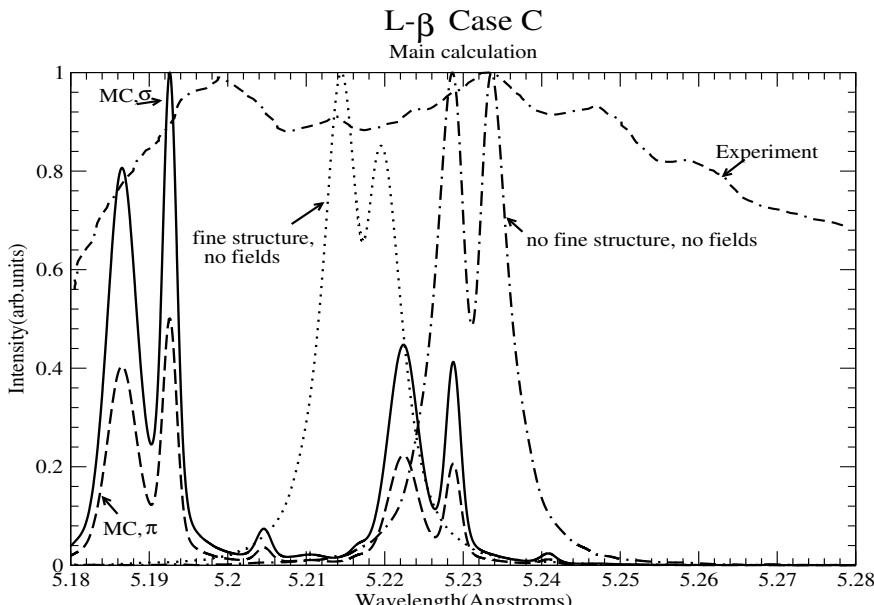

**Figure 19.** Autocorrelation functions C(t) for the various components (corresponding to different Blokhintsev–Floquet peaks) of the $L_\beta$, case B calculation. The solid lines correspond to the "main calculation" components; the dotted lines correspond to the two finely structured components in the absence of oscillatory and static fields. The dashed−dotted line corresponds to the calculation without fine structure and without oscillatory or static fields. The dashed lines are the components in the calculation where dressing was turned off.

**Figure 20.** Main calculation for $Ly - \beta$, case C. Shown are the $\sigma$ (solid line) and $\pi$ (dashed line) profiles for the main calculation; the profiles for the calculations without an oscillatory or a static field with (dotted line) or without (dashed−dotted line) fine structure; and the experimental profile (dot−double−dashed line) of [9,10].

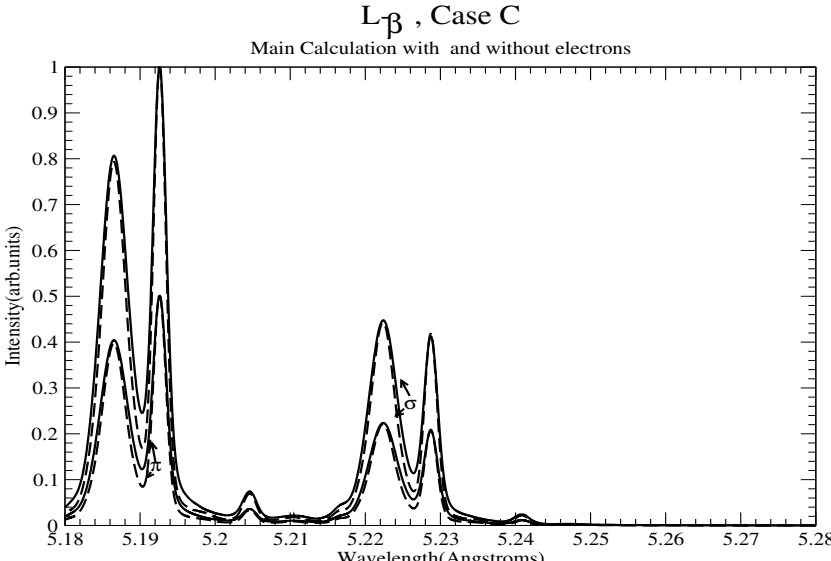

**Figure 21.** Main calculation for $Ly - \beta$, case C with only ion broadening accounted for (i.e., no electrons). Shown are the main calculation's $\sigma$ and $\pi$ profiles (solid line), along with the profiles for the calculations without electron broadening (dashed for the $\sigma$ and dash$-$dot for the $\pi$ profile).

### 4.3.2. Effect of Dressing

As discussed above, the effect of dressing is investigated here also in Figure 22: In addition to the main calculation (solid line), a "no dressing" calculation (dashed line) is shown, where the matrix *S* which described the satellite positions and intensities was correctly computed, but the emitter–plasma interaction in the Schrödinger equation was (incorrectly) not dressed by the nonrandom static and oscillatory fields.

As expected, this results in a broader line, but this effect is completely inadequate to match the profile (or the calculations of Refs. [9,10]).

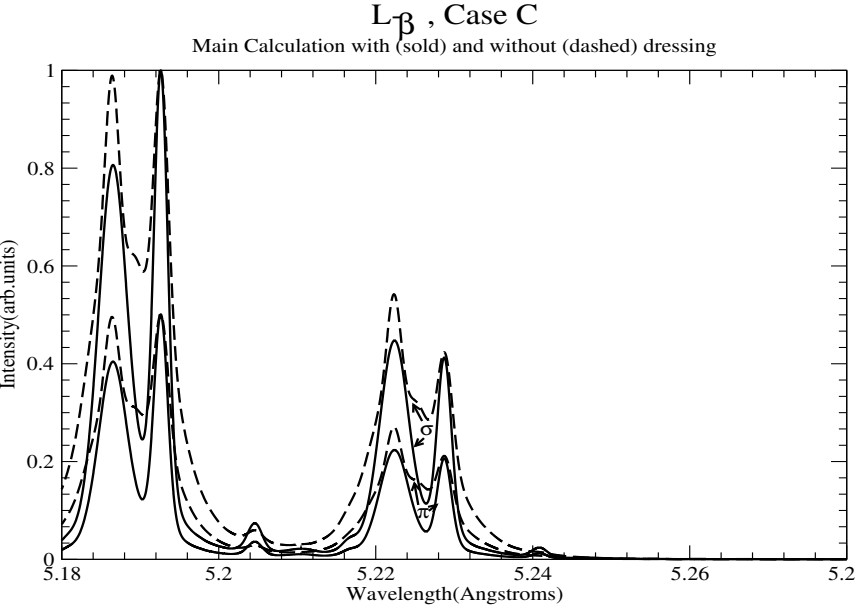

**Figure 22.** Effect of Dressing for $Ly - \beta$, case C. Shown are $\sigma$ and $\pi$ profiles for the main calculation (solid line) and the "no dressing" calculation (dashed line), i.e., the matrix *S* which described the satellite positions and intensities is correctly computed, but the emitter–plasma interaction is (incorrectly) not dressed by the nonrandom static and oscillatory fields.

### 4.3.3. Directional Dependence

Figures 23 and 24 compare the MC to calculations with the same static field—that is, (a) perpendicular and (b) parallel to the static field for the $\sigma$ and $\pi$ components, respectively. While the differences are minor for (a), they are substantial for (b). The closeness of the main and perpendicular profiles is due to the fact that the highest-intensity components are very similar in intensity and position.

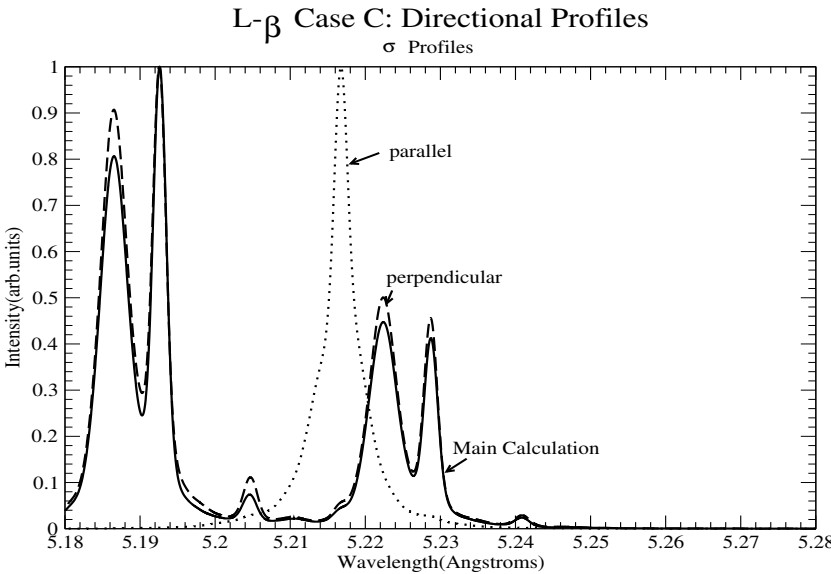

**Figure 23.** Directional Effects for $Ly - \beta$, case C. Shown are $\sigma$ profiles for the main calculation (solid line), and with the entire static field (4.9 GV/cm) when parallel (dotted line) or perpendicular (dashed line) to the oscillatory field.

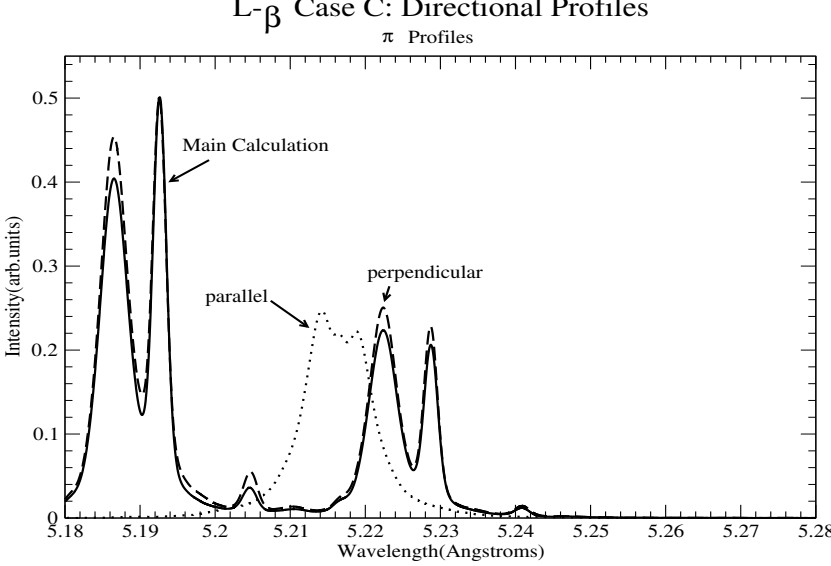

**Figure 24.** Directional effects for $Ly - \beta$, case C. Shown are $\pi$ profiles for the main calculation (solid line) and the entire static field (4.9 GV/cm) when parallel (dotted line) or perpendicular (dashed line) to the oscillatory field.

### 4.3.4. Effect of Randomness of the Static Field

Two calculations are shown in Figures 25 and 26: In both calculations, the static component parallel to the oscillatory field was 3 GV/cm, but in the first, labeled '0 field', the perpendicular component was 0, and in the second, labeled 'x2', it was taken to be 7.74 GV/cm, twice the value in the main calculation. The $\sigma$ and $\pi$ profiles are displayed

separately. The calculations show that the profile—in particular, the satellite intensity—is very sensitive to the static field. Therefore, if the experimental profile is to be attributed to a combination of an oscillatory and a static field, as proposed in Refs. [9,10], then two things are clear: first, a broad distribution of static field intensities would be required, and second, any "dips" would critically depend on the details of that distribution and hence by themselves (without the distribution) cannot provide any reliable diagnostics.

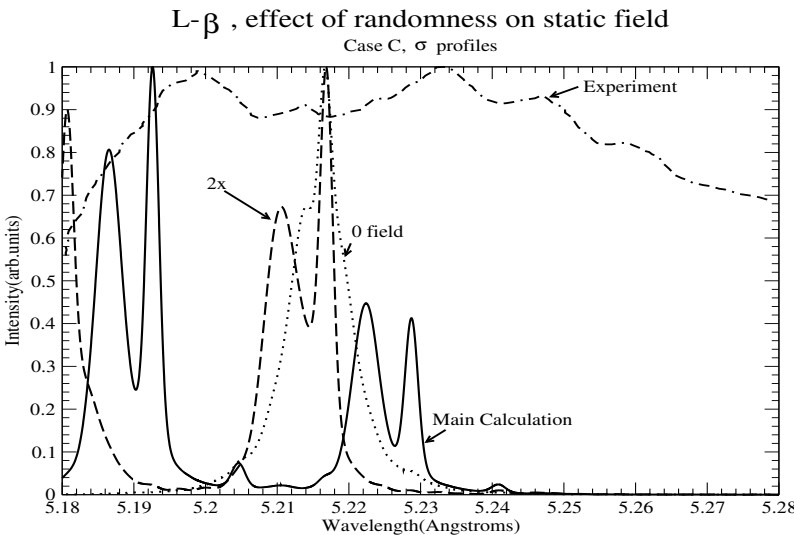

**Figure 25.** Randomness of the static component perpendicular to the oscillatory field for $Ly - \beta$, case C. Shown are $\sigma$ profiles for the main calculation (solid line), the component perpendicular to oscillatory field 0 (0 field-dotted) and the component perpendicular to the oscillatory field but twice the size of that in the main calculation, i.e., 7.74 GV/cm (2x-dashed). The static component perpendicular to the oscillatory field was the same as in the main calculation, 3 GV/cm. Shown as well is the experimental profile (dashe−dotted line).

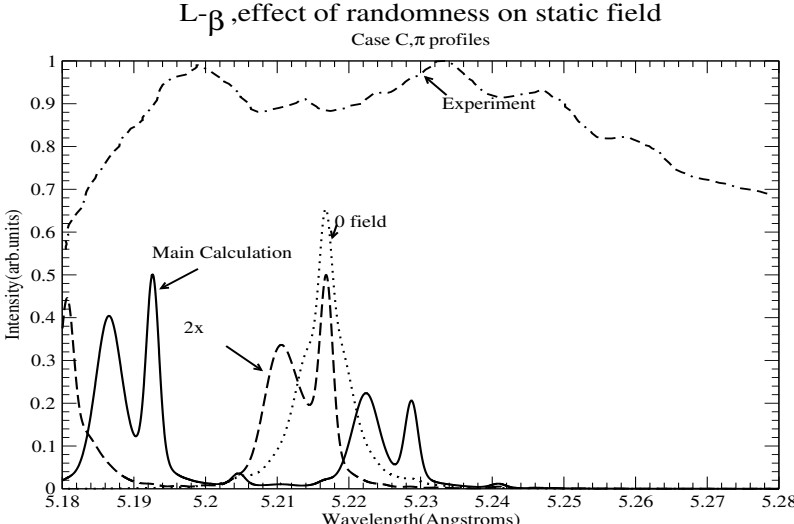

**Figure 26.** Randomness of static component perpendicular to the oscillatory field for $Ly - \beta$, case C. Shown are $\pi$ profiles for the main calculation (solid line), the component perpendicular to oscillatory field 0 (0 field-dotted) and the component perpendicular to the oscillatory field but twice the size of that in the main calculation, i.e., 7.74 GV/cm (2x-dashed). The static component perpendicular to the oscillatory field was the same as in the main calculation, 3 GV/cm. Shown as well is the experimental profile (dashed−dotted line).

4.3.5. Cold Ions

Figure 27 shows the main calculation repeated with cold (T = 1 eV) ions. All other parameters stayed the same as in the main calculation. Shown are the $\sigma$ (solid line) and $\pi$ (dashed–dotted line) main calculation profiles and the corresponding $\sigma$ (dashed line) and $\pi$ (dotted line) profiles for an ion temperature of 1 eV. As expected, the regions between the peaks are filled less and components are resolved, hence exacerbating the differences from the experiment.

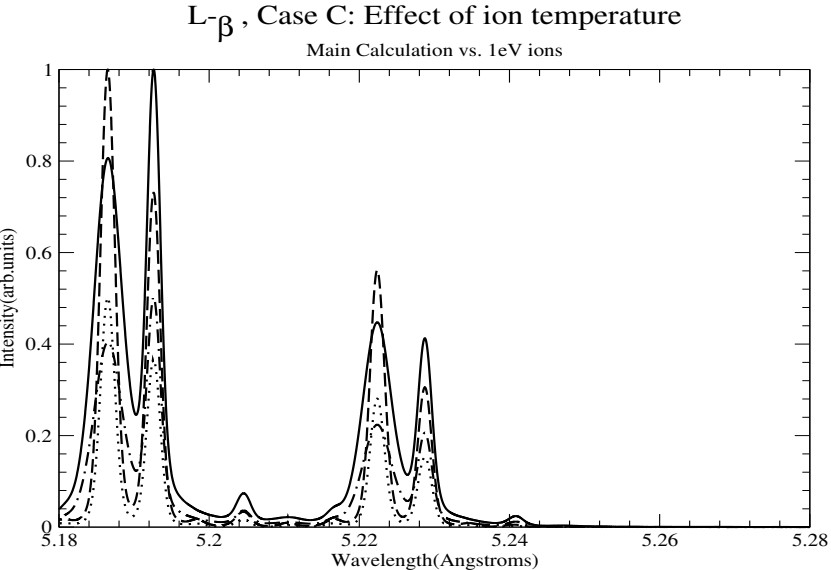

**Figure 27.** $\sigma$ (solid line) and $\pi$ (dashed–dotted line) main calculation profiles and the corresponding $\sigma$ (dashed line) and $\pi$ (dotted line) profiles for an ion temperature of 1 eV.

4.3.6. Static Timescale

Figure 28 shows the autocorrelation functions C(t) of the different components (peaks) for the main calculation. The solid lines correspond to the "main calculation" components, the dotted lines correspond to the two finely structured components in the absence of oscillatory and static fields. The dashed–dotted line corresponds to the calculations without fine structure and without oscillatory or static fields. The dashed lines are the components in the calculation where dressing was turned off. To be static, a field must vary at worst only slowly on the timescale where these C(t) are appreciable; i.e., the variation must be on time scales longer than 3 fs. From Figure 28, we thus infer that turbulence can be considered static if it is associated with frequencies $\ll 0.3 \times 10^{15}$ s$^{-1}$. Again, note that these autocorrelation functions do not become negative (i.e., no dip).

*4.4. Case D*

4.4.1. Main Calculation D

This is a "dip-free" case, according to Refs. [9,10]. Figure 29 displays the results of the main calculation (MC) for the $\sigma$ (solid line) and $\pi$ (dashed line) profiles, for which we took the static field components perpendicular and parallel to the oscillatory field to be 1 GV/cm each. Shown as well are the profiles for the calculations without static or oscillatory fields (i.e., the pure thermal profiles), with (dotted line) and without (dashed–dotted line) fine structure; and the experimental results (dot–double-dashed line) of Refs. [9,10]. Again, the results show that the experimental profile is much wider than that of the main calculation and that satellites (and associated dips) are clearly visible, in contradiction to the method of Refs. [9,10]. As expected, and as may be seen in Figure 30, the profile is determined by ion broadening; electrons only slightly fill in the dips between the various peaks.

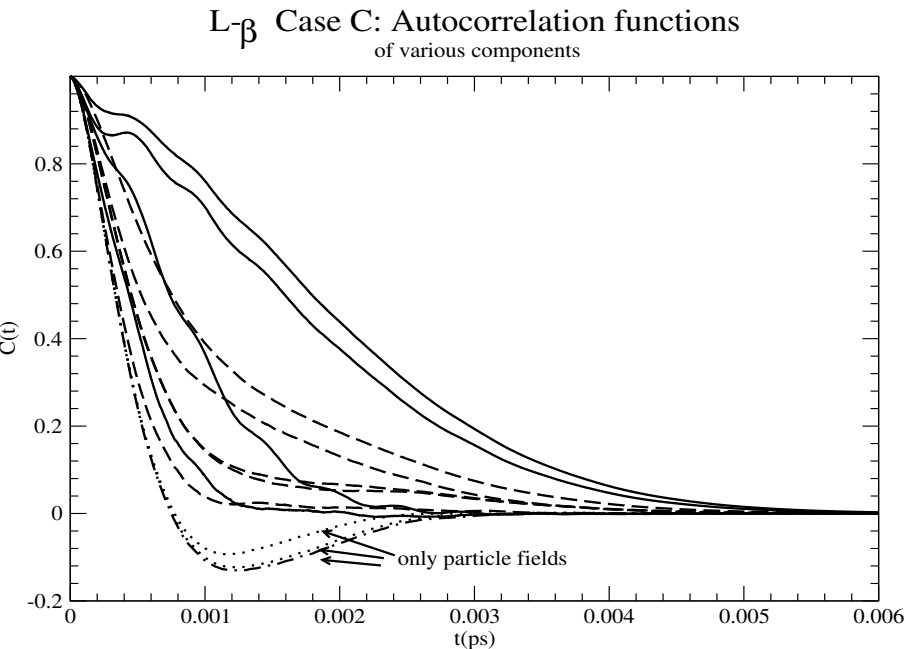

**Figure 28.** Autocorrelation functions C(t) for the various components (corresponding to different Blokhintsev−Floquet peaks) of the $L_\beta$, case C calculation. The solid lines correspond to the "main calculation" components; the dotted lines correspond to the two finely structured components in the absence of oscillatory and static fields. The dashed−dotted line corresponds to the calculation without fine structure and without oscillatory or static fields. The dashed lines are the components of the calculation where dressing was turned off.

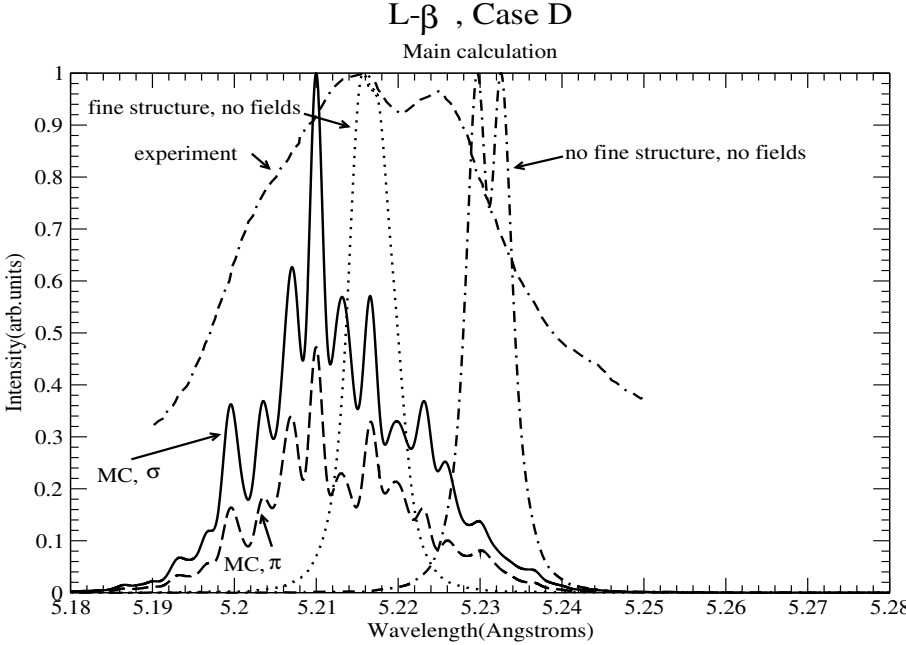

**Figure 29.** Main calculation for $Ly - \beta$, case D. Shown are the $\sigma$ (solid line) and $\pi$ (dashed line) profiles for the main calculation; the profiles for the calculations without an oscillatory or a static field, with (dotted line) and without (dashed–dotted line) fine structure; and the experimental profiles (dot–double-dashed line) of Refs. [9,10].

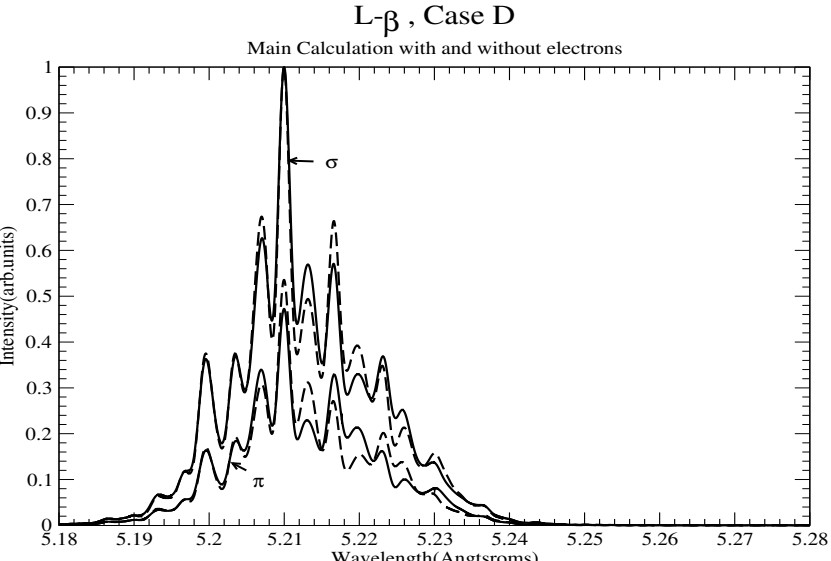

**Figure 30.** Main calculation for $Ly - \beta$, case D, with only ion broadening accounted for (i.e., no electrons). Shown are the main calculation's $\sigma$ and $\pi$ profiles (solid line), along with the profiles for the calculations without electron broadening (dashed for the $\sigma$ and dash–dot for the $\pi$ profile).

### 4.4.2. Effect of Dressing

As discussed above, the effect of dressing is investigated here also in Figure 31: In addition to the main calculation (solid line), a "no dressing" calculation (dashed line) is shown, where the matrix $S$ which described the satellite positions and intensities is correctly computed, but the emitter–plasma interaction in the Schrödinger equation is (incorrectly) not dressed by the nonrandom static and oscillatory fields.

As expected, this results in a broader line, as the components merge, but once again, this is completely inadequate to match the profile (or the calculations of Refs. [9,10].

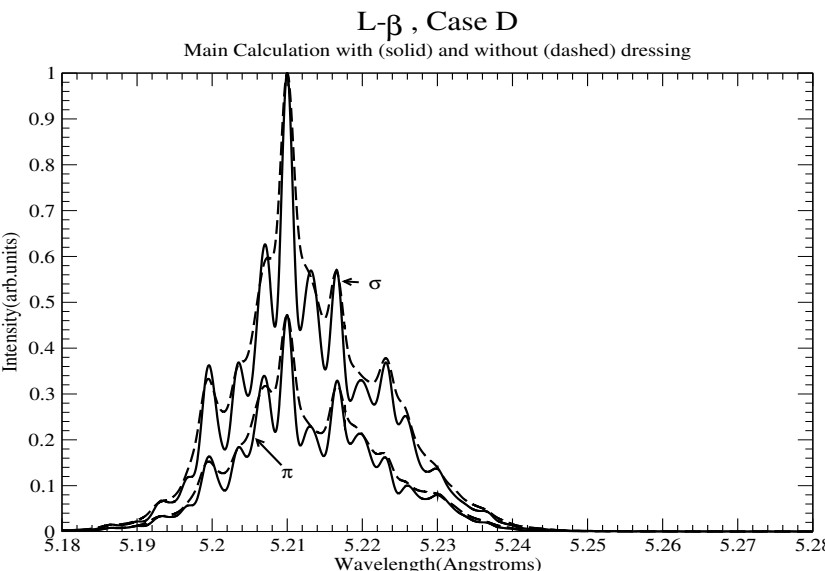

**Figure 31.** Effect of dressing for $Ly - \beta$, case D. Shown are $\sigma$ and $\pi$ profiles for the main calculation (solid line) and the "no dressing" calculation (dashed line); i.e., the matrix $S$ which described the satellite positions and intensities is correctly computed, but the emitter–plasma interaction is (incorrectly) not dressed by the nonrandom static and oscillatory fields.

### 4.4.3. Directional Dependence

By keeping the electron density and temperature fixed as in the main calculation (solid line); assuming identical electron, ion and Doppler temperatures; and assuming a Langmuir nonrandom field in the z-direction, the profile was calculated with a static nonrandom field of 2 GV/cm which (a) is parallel to the Langmuir field (dotted line) and (b) is perpendicular to the Langmuir field (dashed line). It is clear that information on field direction can in principle be extracted from the profiles, though the experiments give no such information. Figures 32 and 33 display the $\sigma$ and $\pi$ profiles respectively.

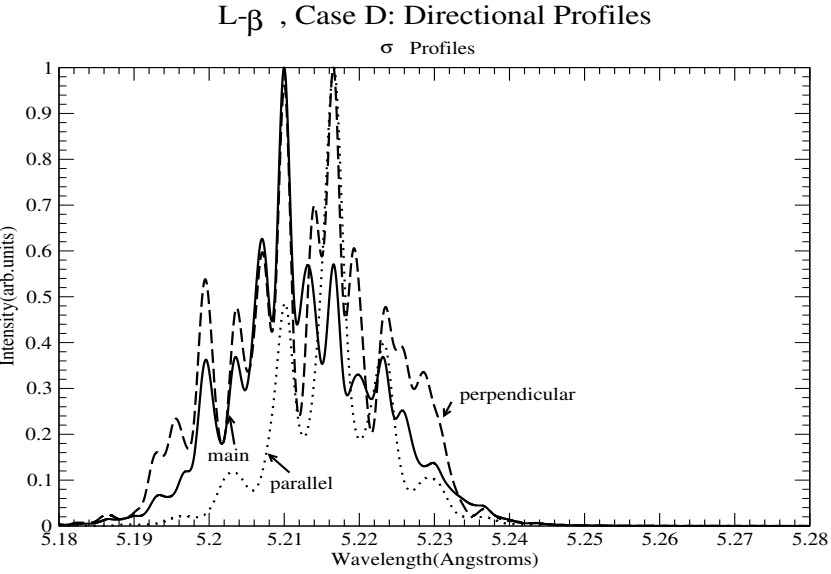

**Figure 32.** Directional Effects for $Ly - \beta$, case D. Shown are $\sigma$ profiles for the main calculation (solid line) and the entire static field (2 GV/cm), parallel (dotted line) or perpendicular (dashed line) to the oscillatory field.

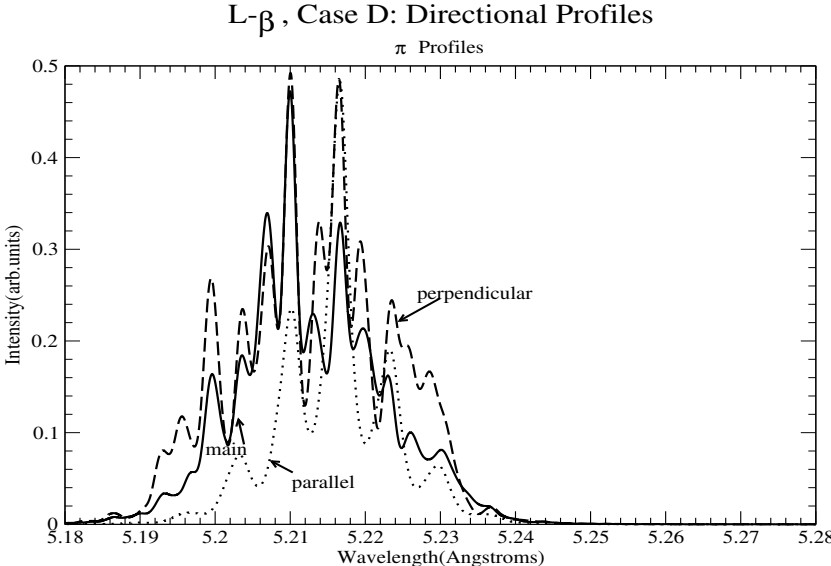

**Figure 33.** Directional effects for $Ly - \beta$, case D. Shown are $\pi$ profiles for the main calculation (solid line) and the entire static field (2 GV/cm) when parallel (dotted line) or perpendicular (dashed line) to the oscillatory field.

### 4.4.4. Effect of Randomness of the Static Field

Two calculations are shown in Figures 34 and 35: In both calculations, the static component parallel to the oscillatory field was 1 GV/cm, but in the first, labeled '0 field',

the perpendicular component was 0, and in the second, labeled 'x2', it was taken to be 2 GV/cm, twice the value of the main calculation. As before, we show the $\sigma$ and $\pi$ profiles separately. The calculations show that even if the distribution functions were to be such as to somehow produce a fairly flat central part, the wing satellites do not have enough intensity to produce a profile that is consistent with the large experimental width.

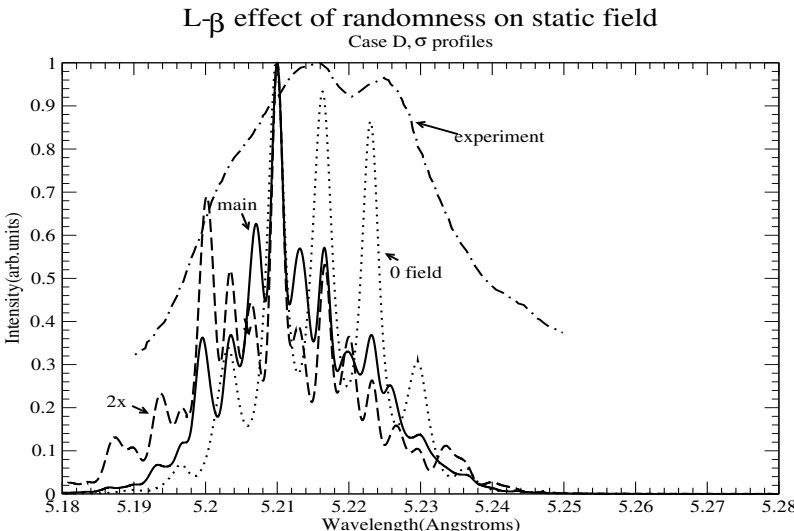

**Figure 34.** Randomness of the static component perpendicular to the oscillatory field for $Ly - \beta$, case D. Shown are $\sigma$ profiles for the main calculation (solid line), the component perpendicular to oscillatory field 0 (0 field-dotted) and the component perpendicular to the oscillatory field but twice the size of that in the main calculation, i.e., 2 GV/cm (2x-dashed). The static component perpendicular to the oscillatory field is the same as in the main calculation, 1 GV/cm. Shown as well is the experimental profile (dashed–dotted line).

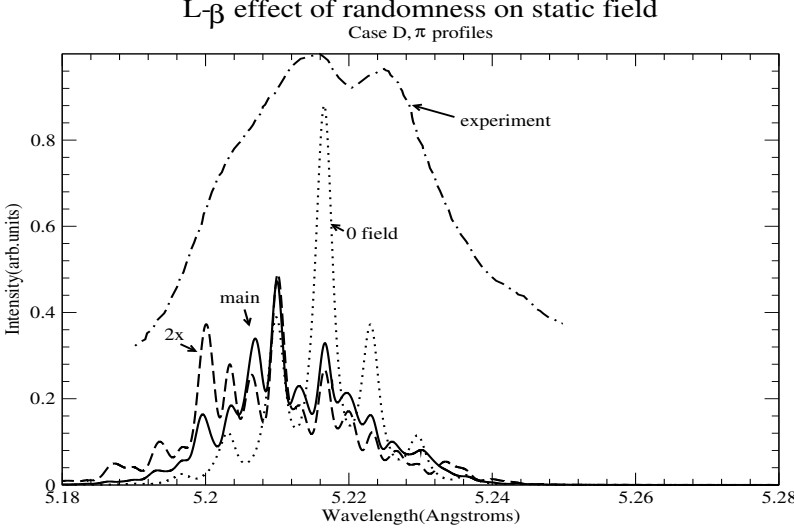

**Figure 35.** Randomness of a static component perpendicular to the oscillatory field for $Ly - \beta$, case D. Shown are $\pi$ profiles for the main calculation (solid line), the component perpendicular to oscillatory field 0 (0 field-dotted) and the component perpendicular to the oscillatory field but twice the size of that in the main calculation, i.e., 2 GV/cm (2x-dashed). The static component perpendicular to the oscillatory field is the same as in the main calculation, 1 GV/cm. Shown as well is the experimental profile (dashed–dotted line).

### 4.4.5. Cold Ions

Figure 36 shows the main calculation repeated with cold (T = 1 eV) ions. All other parameters stayed the same as in the main calculation. Shown are the $\sigma$ (solid line) and $\pi$ (dashed–dotted line) main calculation profiles and the corresponding $\sigma$ (dashed line) and $\pi$ (dotted line) profiles for an ion temperature of 1 eV. As expected, the regions between the peaks are filled less, hence exacerbating the differences from the experiment.

**Figure 36.** $\sigma$ (solid line) and $\pi$ (dashed–dotted line) main calculation profiles and the corresponding $\sigma$ (dashed line) and $\pi$ (dotted line) profiles for an ion temperature of 1 eV.

### 4.4.6. Static Timescale

Figure 37 shows the autocorrelation functions, C(t), of the different components (peaks) for the main calculation. The solid lines correspond to the "main calculation" components, the dotted lines correspond to the two finely structured components in the absence of oscillatory and static fields. The dashed–dotted line corresponds to the calculation without fine structure and without oscillatory or static fields. The dashed lines are the components in the calculation where dressing was turned off. To be static, a field must vary at worst only slowly on the timescale where these C(t) are appreciable; i.e., the variation must be on time scales longer than 3–4 fs. We thus infer that turbulence can be considered static if it is associated with frequencies $\ll 0.3 \times 10^{15}$ s$^{-1}$. Again, in contrast to the normal case (i.e., without a periodic field), where C(t) becomes negative, leading to a central dip in the profile, C(t) in the main calculation does not become negative.

### 4.5. Case E

### 4.5.1. Main Calculation E

Figure 38 displays the results of the main calculation (MC) for the $\sigma$ (solid line) and $\pi$ (dashed line) profiles. The static field was taken to be 3.348 and 2 GV/cm in the x and z-directions (parallel to the oscillatory field), respectively. Shown as well are the results without static and oscillatory fields (i.e., the pure thermal profiles) and with (dotted line) and without (dashed–dotted line) fine structure, and the experimental results (dot–double-dashed line) of Refs. [9,10]. Again, the results show that the experimental profile is much larger than the calculation, and that satellites (and associated dips) are clearly visible, in contradiction to the method of Refs. [9,10]. As expected, and as may be seen in Figure 39 that the profile is determined by ion broadening; electrons only slightly fill in the dips between the various peaks.

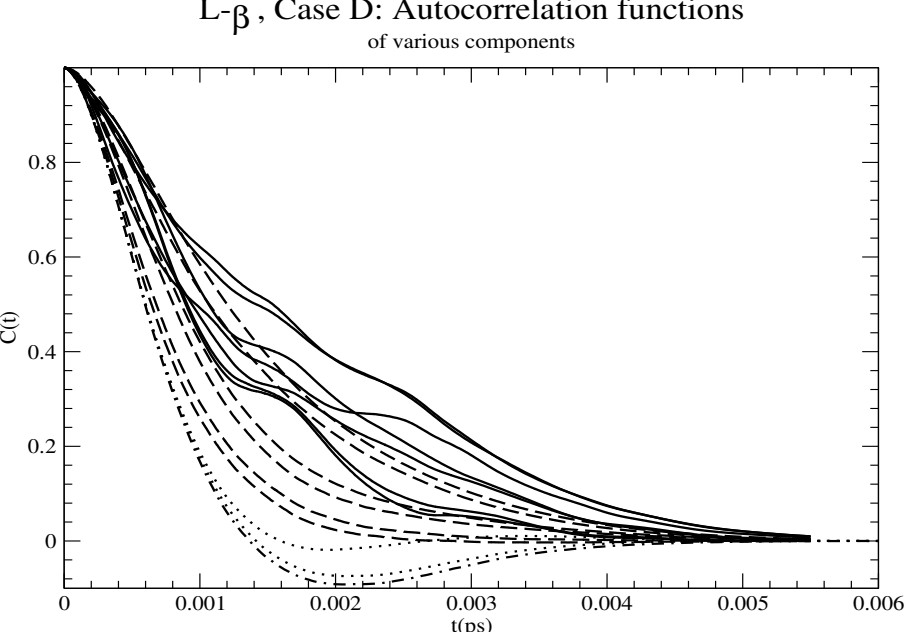

**Figure 37.** Autocorrelation functions C(t) for the various components (corresponding to different Blokhintsev–Floquet peaks) of the $L_\beta$, case D calculation. The solid lines correspond to the "main calculation" components, the dotted lines correspond to the two finely structured components in the absence of oscillatory and static fields. The dashed−dotted line corresponds to the calculation without fine structure and without oscillatory or static fields. The dashed lines are the components in the calculation where dressing was turned off.

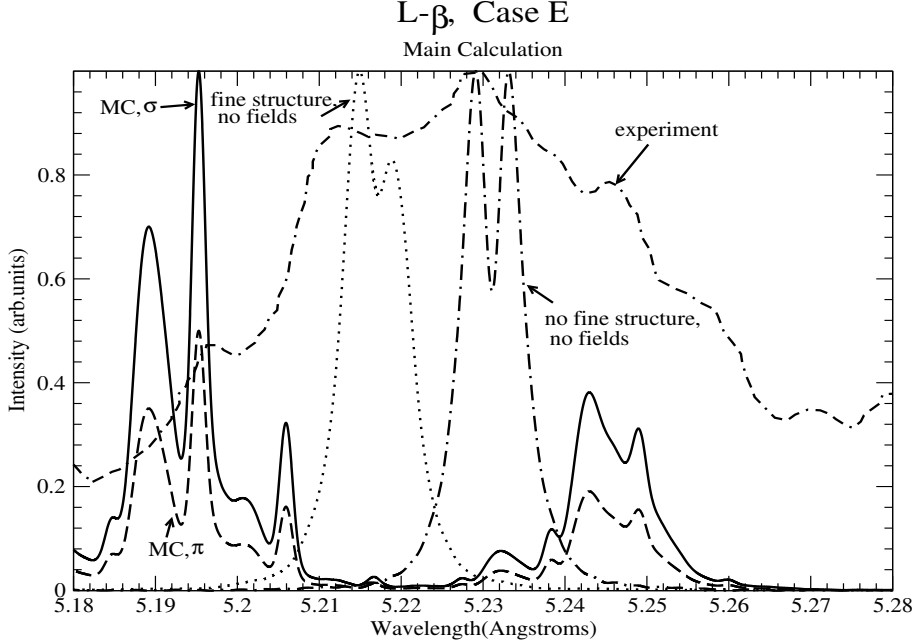

**Figure 38.** Main calculation for $Ly - \beta$, case E. Shown are the $\sigma$ (solid line) and $\pi$ (dashed line) profiles for the main calculation; profiles for the calculations without an oscillatory or static field, with (dotted line) or without (dashed–dotted line) fine structure; and the experimental profile (dot–double-dashed line) of Ref. [10].

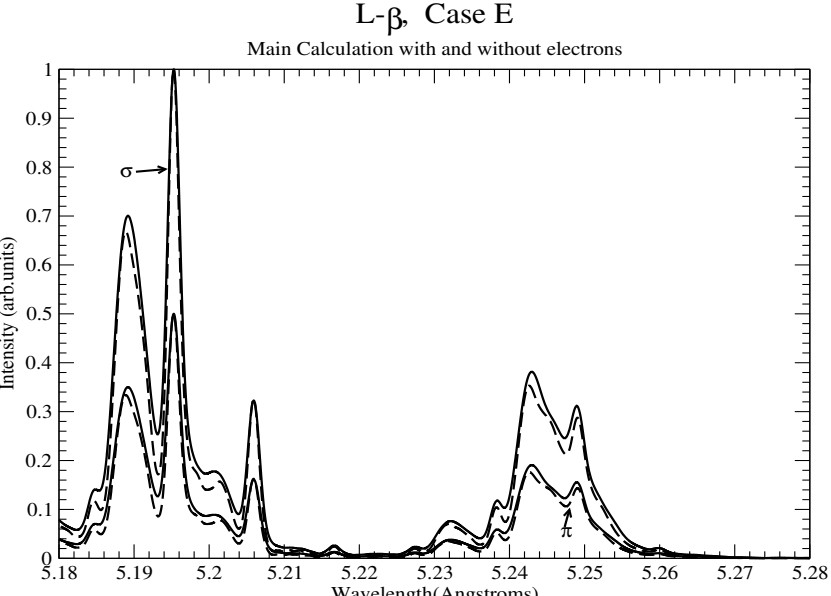

**Figure 39.** Main calculation for $Ly - \beta$, case E with only ion broadening accounted for (i.e., no electrons). Shown are the main calculation's $\sigma$ and $\pi$ profiles (solid line), together with the profiles for the calculations without electron broadening (dashed for the $\sigma$ and dash–dot for the $\pi$ profile).

4.5.2. Effect of Dressing

As discussed above, the effect of dressing is investigated here also, in Figure 40: In addition to the main calculation (solid line), a "no dressing" calculation (dashed line) is shown. The matrix *S* which described the satellite positions and intensities was correctly computed, but the emitter–plasma interaction in the Schrödinger equation was (incorrectly) not dressed by the nonrandom static and oscillatory fields. As expected this results in a broader line, but this effect is completely inadequate to match the profile (or the calculations of Refs. [9,10]).

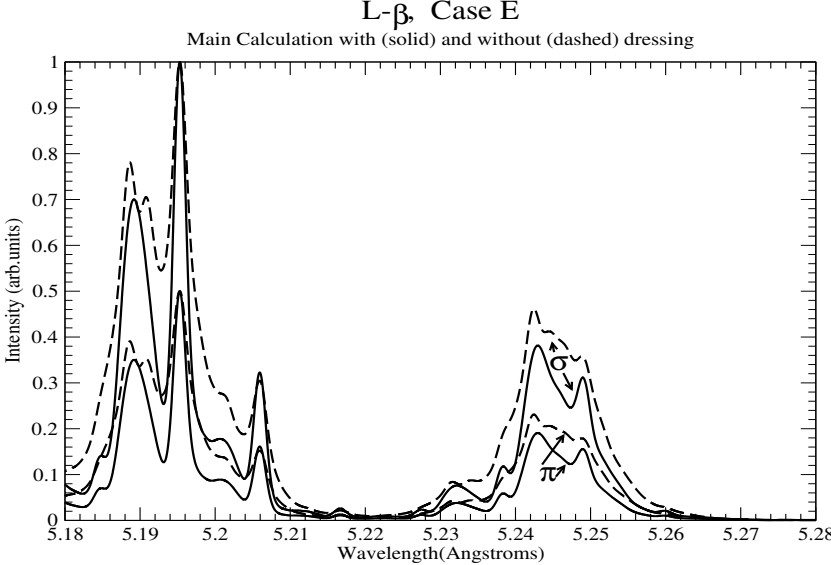

**Figure 40.** Effect of dressing for $Ly - \beta$, case E. Shown are $\sigma$ and $\pi$ profiles for the main calculation (solid line) and the "no dressing" calculation (dashed line); i.e., the matrix *S* which describes the satellite positions and intensities is correctly computed, but the emitter–plasma interaction is (incorrectly) not dressed by the nonrandom static and oscillatory fields.

### 4.5.3. Directional Dependence

Keeping the electron density and temperature fixed, as in the main calculation (solid line), and further assuming identical electron, ion and Doppler temperatures and assuming a Langmuir nonrandom field in the z-direction, the profile was calculated with a static nonrandom field of 3.9 GV/cm, which a) is parallel to the Langmuir field (dotted line) and b) is perpendicular to the Langmuir field (dashed line). Figures 41 and 42 show very different profiles for the parallel case.

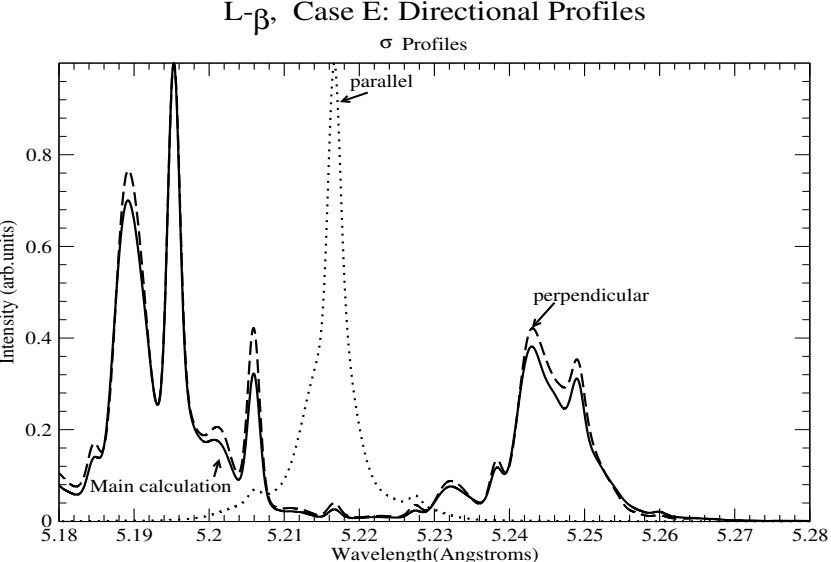

**Figure 41.** Directional effects for $Ly - \beta$, case E. Shown are $\sigma$ profiles for the main calculation (solid line) and the entire static field (3.9 GV/cm) when parallel (dotted line) and perpendicular (dashed line) to the oscillatory field.

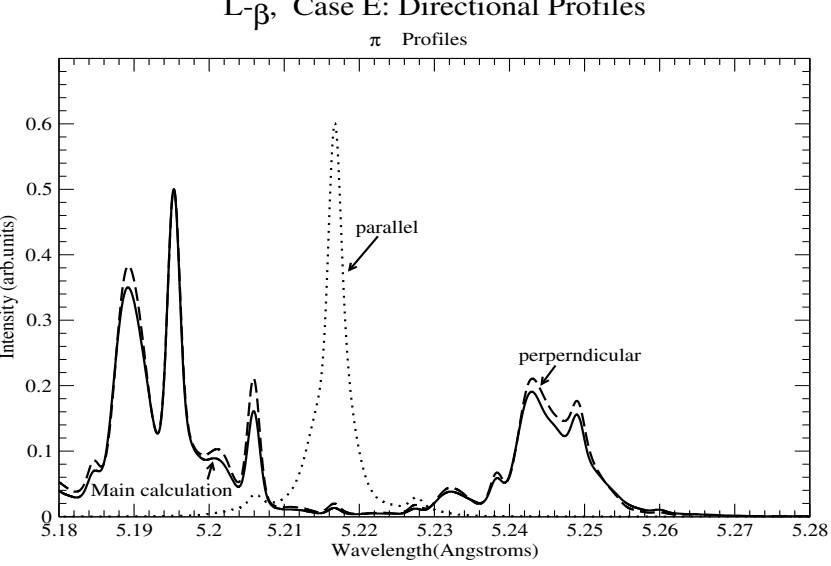

**Figure 42.** Directional effects for $Ly - \beta$, case E. Shown are $\pi$ profiles for the main calculation (solid line) and the entire static field (3.9 GV/cm) when parallel (dotted line) and perpendicular (dashed line) to the oscillatory field.

### 4.5.4. Effect of Randomness of the Static Field

Two calculations are shown in Figures 43 and 44: In both calculations, the static component parallel to the oscillatory field was 2 GV/cm, but in the first, labeled '0 field', the perpendicular component was 0, and in the second, labeled 'x2', it was taken to be

6.696 GV/cm, twice the value of the main calculation. As before, we show the $\sigma$ and $\pi$ profiles separately. Again, even if the distribution functions were such as to somehow produce a fairly flat central part, the wing satellites do not have enough intensity to produce a profile that is consistent with the large experimental width.

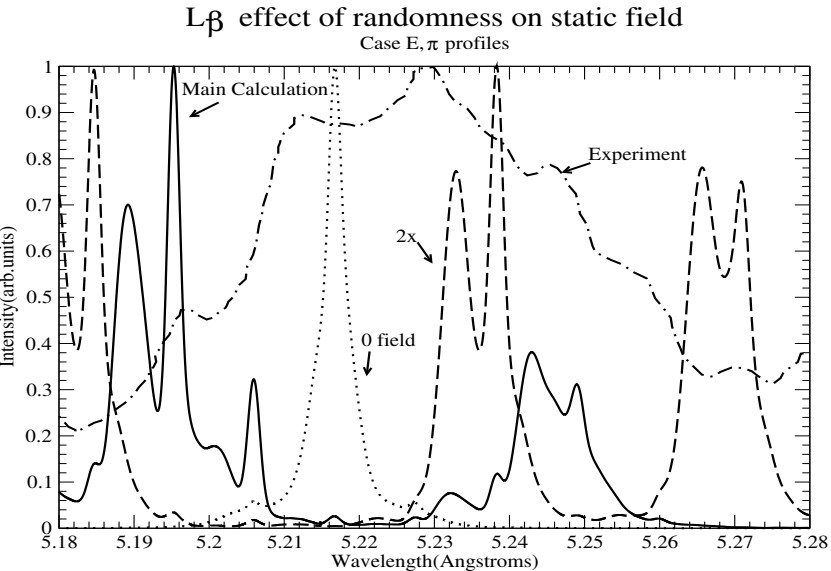

**Figure 43.** Randomness of static component perpendicular to the oscillatory field for $Ly - \beta$, case E. Shown are $\sigma$ profiles for the main calculation (solid line), the component perpendicular to oscillatory field 0 (0 field-dotted) and the component perpendicular to the oscillatory field but twice the size of that in the main calculation, i.e., 6.696 GV/cm (2x-dashed). The static component perpendicular to the oscillatory field is the same as in the main calculation, 2 GV/cm. Shown as well is the experimental profile (dash−dotted line).

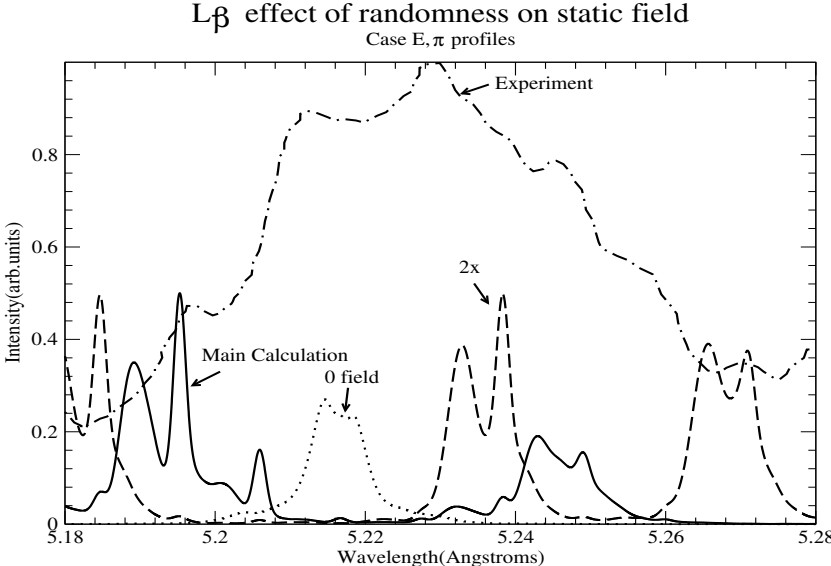

**Figure 44.** Randomness of a static component perpendicular to the oscillatory field for $Ly - \beta$, case E. Shown are $\pi$ profiles for the main calculation (solid line), the component perpendicular to oscillatory field 0 (0 field-dotted) and the component perpendicular to the oscillatory field but twice the size of that in the main calculation, i.e., 6.696 GV/cm (2x-dashed). The static component perpendicular to the oscillatory field was the same as in the main calculation, 2 GV/cm. Shown as well is the experimental profile (dash−dotted line).

### 4.5.5. Cold Ions

Figure 45 shows the main calculation repeated with cold (T = 1 eV) ions. All other parameters stayed the same as in the main calculation. Shown are the $\sigma$ (solid line) and $\pi$ (dashed–dotted line) main calculation profiles and the corresponding $\sigma$ (dashed line) and $\pi$ (dotted line) profiles for an ion temperature of 1 eV. As expected, the regions between the peaks are filled less, hence exacerbating the differences from the experiments.

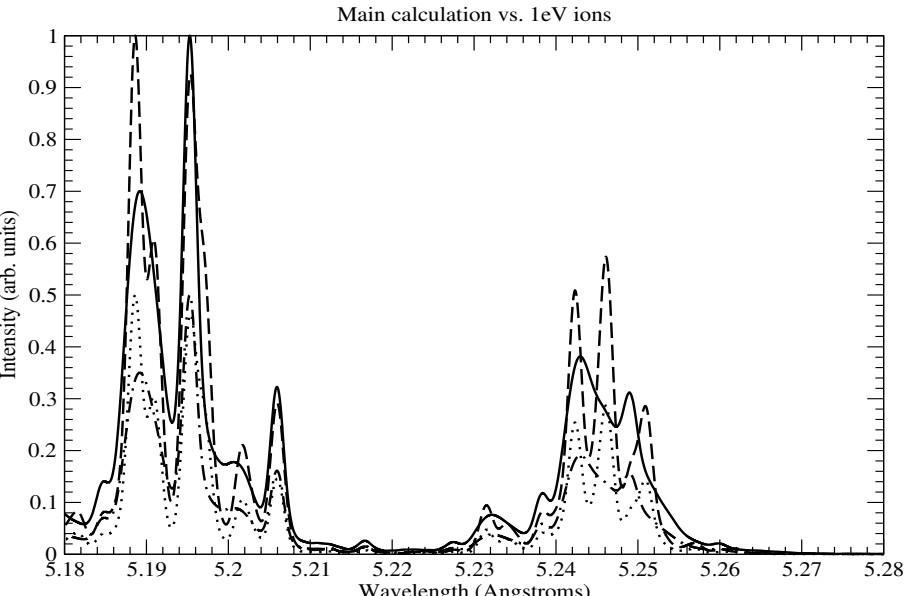

**Figure 45.** $\sigma$ (solid line) and $\pi$ (dashed–dotted line) main calculation profiles and the corresponding $\sigma$ (dashed line) and $\pi$ (dotted line) profiles for an ion temperature of 1 eV.

### 4.5.6. Static Timescale

Figure 46 shows the autocorrelation functions, C(t), of the different components (peaks) for the main calculation. To be static, a field must vary at worst only slowly on the timescale where these C(t) are appreciable; i.e., the variation must be on time scales longer than 1–4 fs. We thus infer that turbulence can be considered static if it is associated with frequencies $\ll \times 10^{15} \mathrm{s}^{-1}$. Note that in contrast to the normal case (i.e., without a periodic field), where C(t) becomes negative, leading to a central dip in the profile, here, C(t) does not become negative. This is in contrast to the decay of C(t) in fine structural components (the two bold lines in the graph) in the absence of oscillatory or static fields, where the decay is much faster and a dip (negative region) is clearly visible. Again, the dressing by the periodic field results in a significantly slower decay of C(t), and hence line narrowing [20–23].

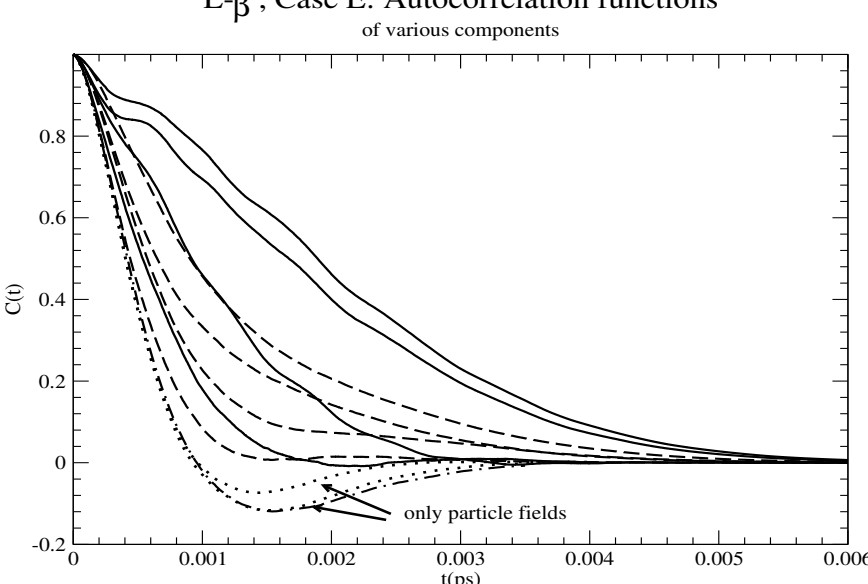

**Figure 46.** Autocorrelation functions, C(t), for the various components (corresponding to different Blokhintsev−Floquet peaks) of the $L_\beta$, case E calculation. Shown as well (dashed−dotted line) are the autocorrelation function for the calculation without static or oscillatory fields and with no fine structure, and the two $Ly - \beta$ components with fine structure but without any oscillatory or static fields (dotted line), i.e., normal calculations. The dashed lines display the corresponding autocorrelation functions without dressing.

## 5. $Ly - \gamma$ Calculations

Ref. [10] provides details for a single $Ly - \gamma$ measurement, namely, electron density $3.6 \times 10^{22}$e/cm$^3$, T = 500 eV (in all calculations labeled "main", electron, ion and Doppler temperatures are assumed equal) and a linearly polarised (Langmuir) wave $E_0 cos(\omega_p t)$ with $\omega_p$ the electron plasma frequency (about $10^{16}$s$^{-1}$) corresponding to the stated electron density (in fact the electron density is inferred from the dip positions) and $E_0 = 0.6$ GV/cm. In addition, a static field **F** , not collinear with the Langmuir field with a root-mean square amplitude of 2.1 GV/cm, was invoked. These parameters constitute what we call the "main calculation" and should be understood for all $Ly - \gamma$ calculations unless otherwise stated. In the main calculation shown in Figure 47, F has an x-component of 1.85 GV/cm and a z-component of 1 GV/cm. These parameters are summarized in Table 2.

**Table 2.** *Ly-γ* calculation parameters.

| El.Density ($\times 10^{21}$e/cm$^3$) | T (eV) | F (GV/cm) | $E_0$ (GV/cm) | Ref. |
|:---:|:---:|:---:|:---:|:---:|
| 36 | 500 | 2.1 | 0.6 | Figure 3A in [10] |

### 5.1. Main Calculation

Figure 47 displays the results of the main calculation (MC) for the $\sigma$ (solid line) and $\pi$ (dashed line) profiles. In the calculations, we used a static nonrandom field of 2.1 GV/cm with a 1 GV/cm component parallel to the Langmuir field and a 1.85 GV/cm component perpendicular to it. Shown as well are the results without static or oscillatory fields (i.e., the pure thermal profiles) with (dotted line) and without (dashed–dotted line) fine structure, and the experimental results (dot–double-dashed line) of Refs. [9,10]. Again, the results show that the experimental profile is much wider than the calculations, and that satellites (and associated dips) are clearly visible, in contradiction to the method of Refs. [9,10]. It is clear that this profile has no resemblance to the profile shown in Ref. [10], which is much broader.

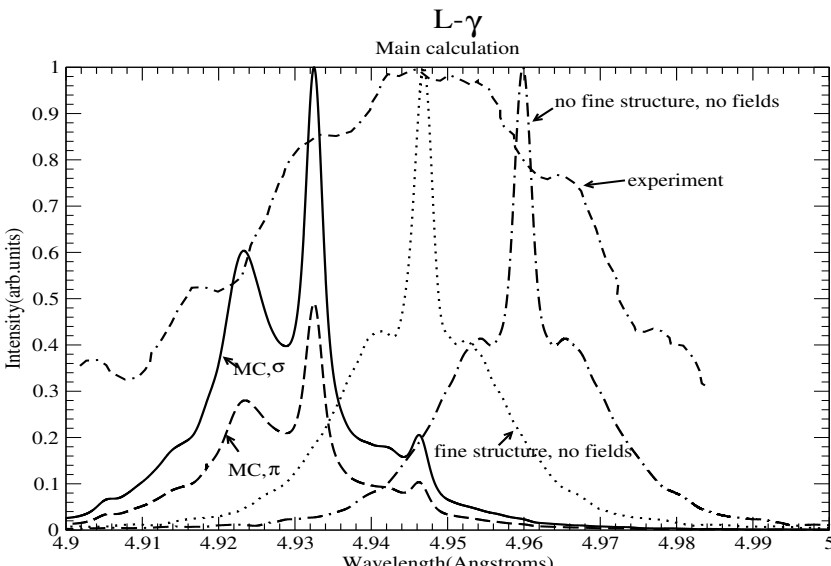

**Figure 47.** Main calculation for $Ly - \gamma$. Shown are the $\sigma$ (solid line) and $\pi$ (dashed line) profiles for the main calculation, together with the profile without an oscillatory and static field with (dotted line) and without (dashed−dotted line) fine structure and the experimental profile (dot−double-dashed line) of Refs. [9,10].

Figure 48 compares the MC with a corresponding calculation neglecting electron broadening and demonstrates that electron broadening makes a substantial contribution, especially for the regions between the peaks.

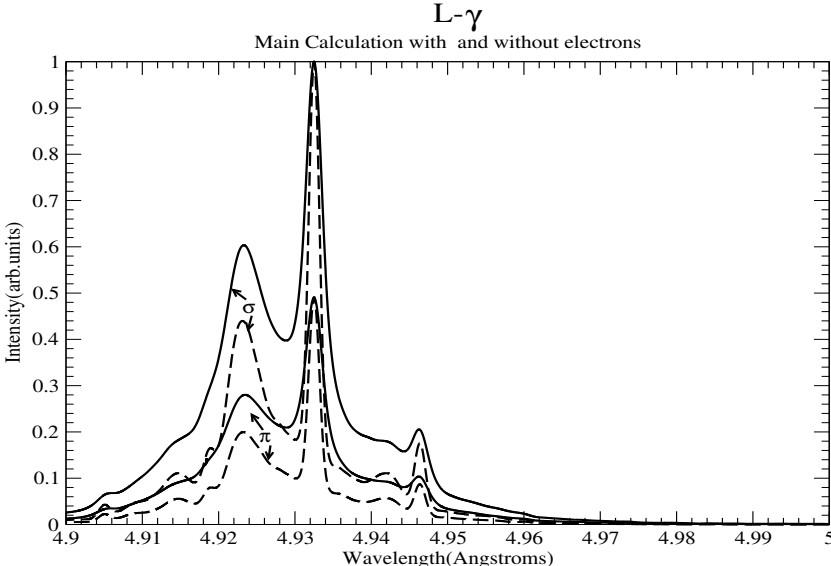

**Figure 48.** Main calculation for $Ly - \gamma$ with only ion broadening accounted for (i.e., no electrons). Shown are the main calculation $\sigma$ and $\pi$ profiles (solid line), together with the profiles for the calculations without electron broadening (dashed).

## 5.2. Effect of Dressing

As discussed above, the effect of dressing is investigated in Figure 49: In addition to the main calculation (solid line), a "no dressing" calculation (dashed line) is shown, where the matrix $S$ which described the satellite positions and intensities is correctly computed, but the emitter–plasma interaction in the Schrödinger equation is (incorrectly) not dressed by the nonrandom static and oscillatory fields. Once again, this results in a broader line, but this effect is completely inadequate to match the profile (or the calculations of Refs. [9,10]).

In other words, as for the $Ly - \beta$ case, the results of these references cannot be attributed to the possible neglect of dressing.

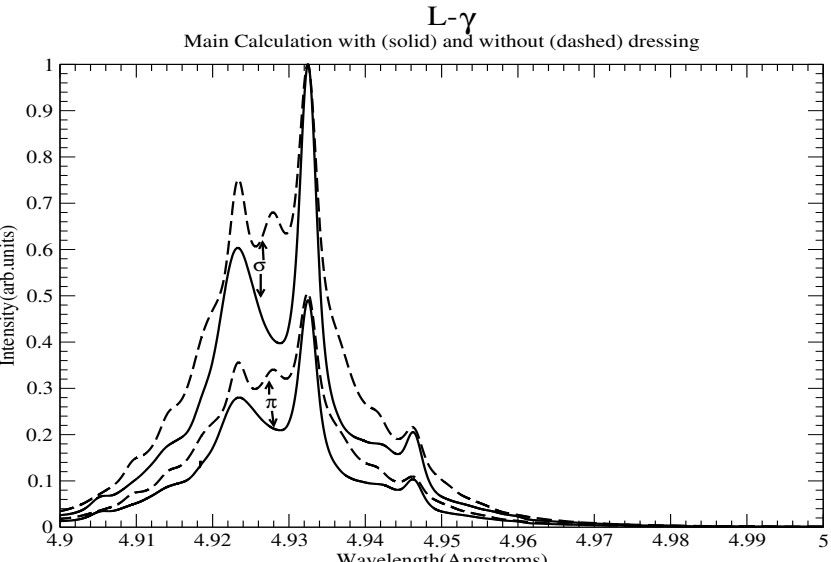

**Figure 49.** Shown are $\sigma$ and $\pi$ profiles for the main calculation (solid line) and the "no dressing" calculation (dashed line) for $Ly - \gamma$.

### 5.3. Directional Dependence

As in the main calculation (solid line), in Figures 50 and 51, we keep the electron density and temperature fixed and further assume identical electron, ion and Doppler temperatures and assume a Langmuir nonrandom field in the z-direction. The profile was calculated with a static nonrandom field of 2.1 GV/cm which was (a) parallel to the Langmuir field (dotted line) and (b) perpendicular to the Langmuir field (dashed line). Again, the profiles and the dips in particular are sensitive to these changes.

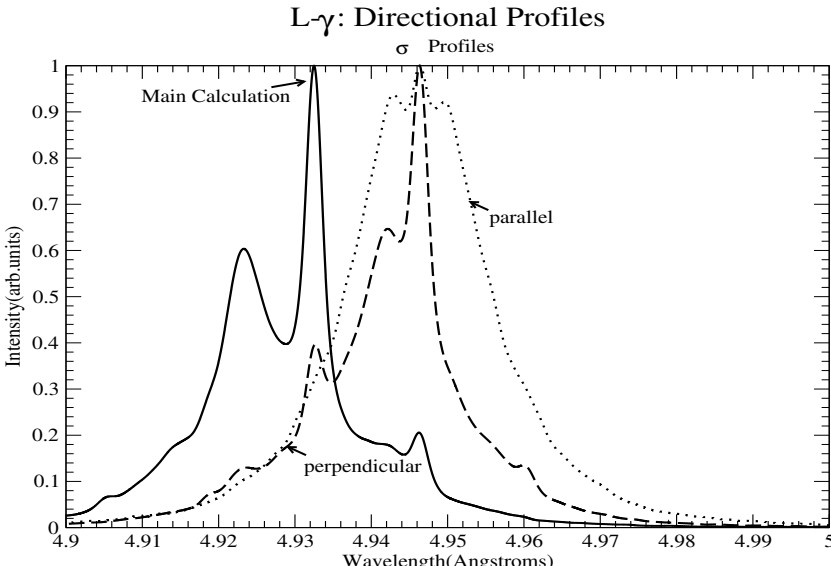

**Figure 50.** Directional Effects for $Ly - \gamma$. Shown are $\sigma$ profiles for the main calculation (solid line) and the entire static field (2.1 GV/cm) when parallel (dotted line) or perpendicular (dashed line) to the oscillatory field.

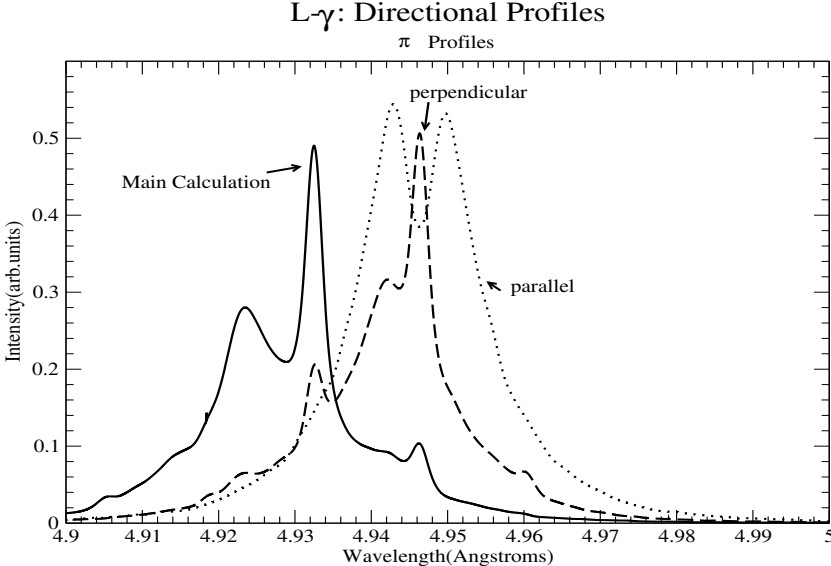

**Figure 51.** Directional Effects for $Ly - \gamma$. Shown are $\pi$ profiles for the main calculation (solid line) and the entire static field (2.1 GV/cm) when parallel (dotted line) or perpendicular (dashed line) to the oscillatory field.

### 5.4. Effect of Randomness of Static Field

If the oscillatory field is not random, then randomness in the static field **F** will in principle shift the positions of the Floquet exponents and thus broaden the line. To access this, calculations were repeated for Figures 52 and 53 with an x-component of the static field equal to a) 0 or b) 3.9 GV/cm, i.e., double the size of the x-component in the main calculation. The results suggest that this is still inadequate to explain the broad feature observed in [10]. Moreover, the dip positions differ greatly for different values of the x-component of the static field, and hence, once again cannot be used as a diagnostic.

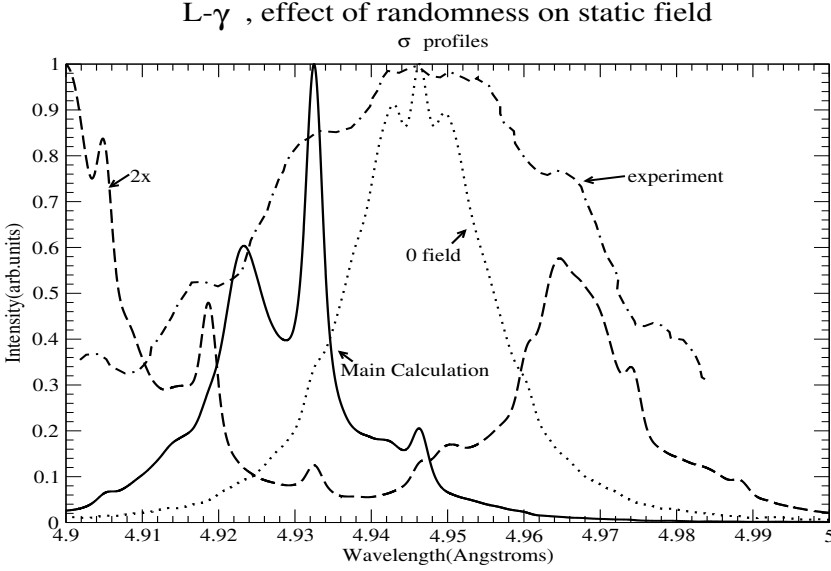

**Figure 52.** Randomness of static component perpendicular to the oscillatory field for $Ly - \gamma$. Shown are $\sigma$ profiles for the main calculation (solid line), the component perpendicular to oscillatory field 0 (0 field-dotted) and the component perpendicular to the oscillatory field but twice the size of that in the main calculation, i.e., 3.9 GV/cm (2x-dashed). The static component perpendicular to the oscillatory field is the same as in the main calculation, 1 GV/cm. Shown as well is the experimental profile (dashed–dotted line).

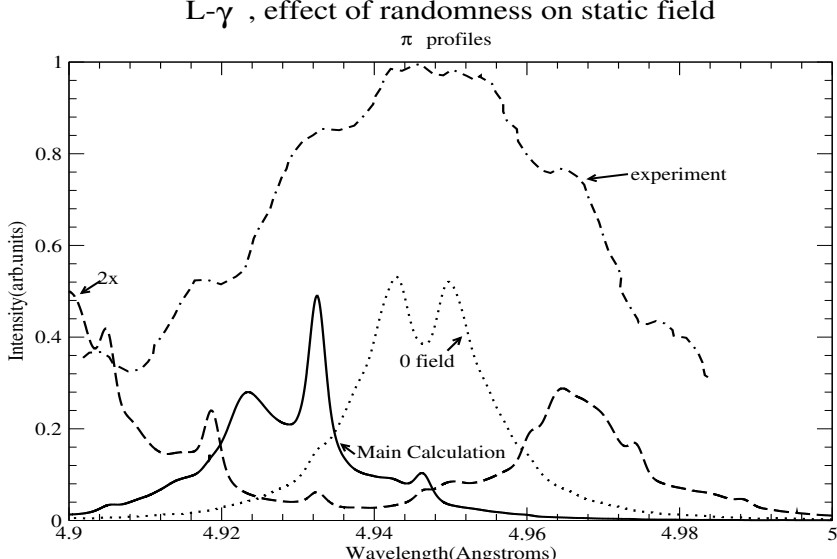

**Figure 53.** Randomness of the static component perpendicular to the oscillatory field for $Ly - \gamma$. Shown are $\pi$ profiles for the main calculation (solid line), for the component perpendicular to oscillatory field 0 (0 field-dotted) and for the component perpendicular to the oscillatory field but twice the size of that in the main calculation, i.e., 3.9V/cm (2x-dashed). The static component perpendicular to the oscillatory field is the same as in the main calculation: 1 GV/cm. Shown as well is the experimental profile (dashed–dotted line).

*5.5. Cold Ions*

Figure 54 shows the main calculation repeated with cold (T = 1 eV) ions. All other parameters stayed the same as in the main calculation. Shown are the $\sigma$ (solid line) and $\pi$ (dashed–dotted line) main calculation profiles and the corresponding $\sigma$ (dashed line) and $\pi$ (dotted line) profiles for an ion temperature of 1 eV. As expected, the regions between the peaks are filled less, hence exacerbating the differences from the experiment, and the effect, as expected, is much stronger than the corresponding differences found in the $Ly - \beta$ case. In addition, components that were merged are now separated, and new dips appear. Hence, due to its effect on broadening, temperature is important and dip positions are not independent of it.

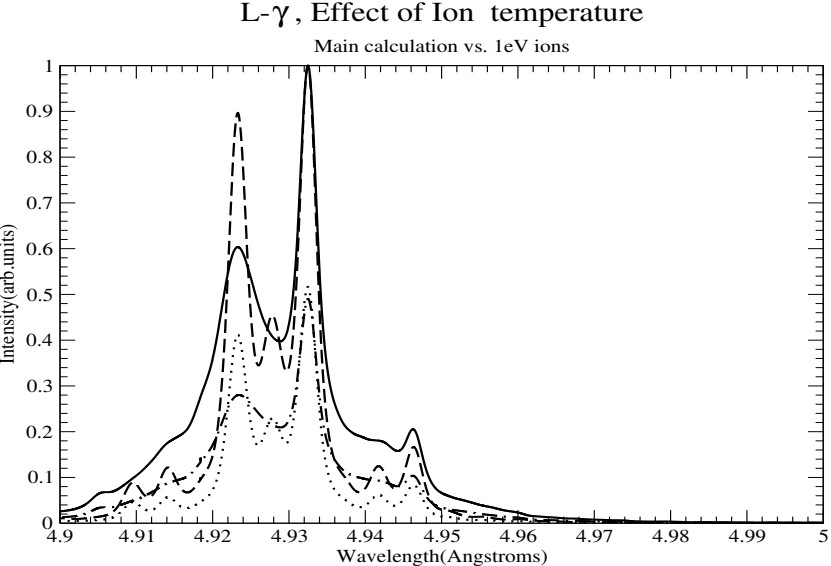

**Figure 54.** $\sigma$ (solid line) and $\pi$ (dashed–dotted line) main calculation profiles and the corresponding $\sigma$ (dashed line) and $\pi$ (dotted line) profiles for an ion temperature of 1 eV.

*5.6. Static Timescales*

Figure 55, shows the autocorrelation functions of the $Ly - \gamma$ components (i.e., Floquet-Blokhintsev peak positions), which are displayed as solid lines. It may be seen that the "lifetime" is in the order of 3fs, and hence the static time scale (whether it refers to ions or ion acoustic waves) is $\ll 0.3 \times 10^{15}$ s$^{-1}$. All components are scaled so that C(0) = 1. Shown as well (dashed–dotted line) is the autocorrelation function for the calculation without static or oscillatory fields and no fine structure, along with the two $Ly - \gamma$ components with a fine structure, but without any oscillatory or static fields (dotted line), i.e., a normal calculation. The dashed lines display the corresponding autocorrelation functions without dressing.

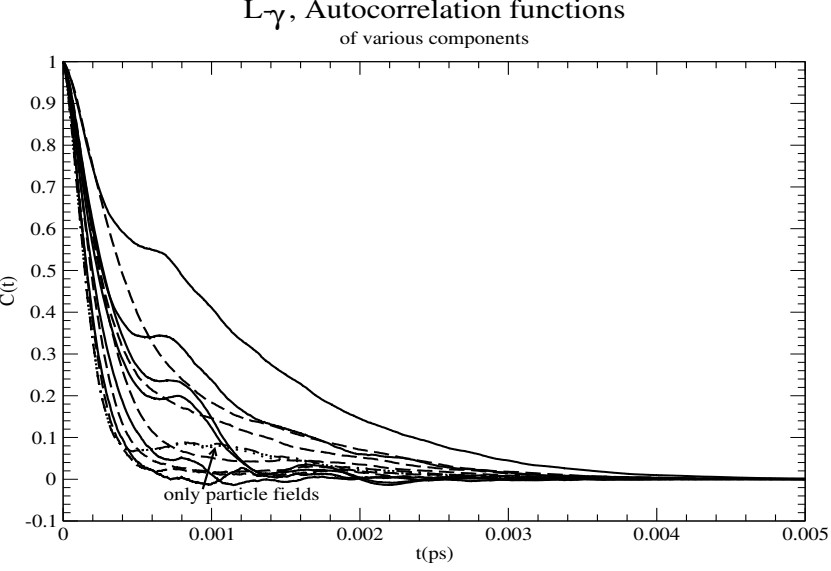

**Figure 55.** C(t) for the strongest $Ly - \gamma$ components (i.e., combinations of Floquet and Blokhintsev positions).

## 6. Conclusions

We computed exactly the line profiles in Refs. [9,10], without assuming quasistatic particle fields for the parameters inferred in these references. It was shown that the profiles bear little or no resemblance to the experimental profiles shown in [9,10]. Hence, if the parameters inferred are correct, the oscillatory and/or static fields invoked must have some randomness; i.e., different emitters see a different time history of these fields. It is then shown that the dip positions are sensitive to the direction and magnitude of the static field alone, along with the temperature. As a result, one cannot use them to diagnose the plasma and/or the invoked turbulent fields. Since there is no corroborating evidence in these references for turbulent fields, as discussed in the analysis of the experiments, it is by no means certain or imperative that the explanation given in these references is correct, and a number of possibilities were mentioned in the analysis of the experiments.

Nevertheless, the prospect of inferring turbulent fields from the spectra is obviously a very interesting question. First, we might expect some kind of directional dependence, and this should be checked experimentally. Second, satellites, if clearly seen, as in the calculations, are a very solid indicator. If the turbulent fields are random with some unknown a priori distribution, how can we get reliable information of these fields and possibly their distribution? If we have a distribution of turbulent fields, each turbulent field realization will, as shown in the calculations, result in peaks (satellites) that are shifted and that have different intensities. Certainly, we can try different distributions until we get a match, but uniqueness is far from guaranteed. The present approach represents a clear, albeit complicated, path to deal with this problem. Some key points to remember are that (a) dressing applies to all random fields, including static random fields; (b) especially in view of the narrowing of the profiles, static field assumptions should be checked, at

least *a posteriori*; and (c) in a correct treatment, "dips" are simply the regions between the peaks. There are no "dip" features in the sense of either a zero profile or a similar depression that is insensitive to the details of the fields, such as directionality and size of the static component.

**Funding:** This research received no external funding.

**Data Availability Statement:** Data is contained within the article.

**Acknowledgments:** Discussions with the Spectral Line Shapes Workshop participants are gratefully acknowledged.

**Conflicts of Interest:** The author declares no conflict of interest.

## Note

[1]    In the present work, the plasma particles are assumed to move in classical trajectories unaffected by the oscillatory and static fields.

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
