# Peer review of "Analysis of Plasma Emission Experiments and ‘Dips’"

_atoms, doi:10.3390/atoms11020029_

Round 1

Reviewer 1 Report

Report on the paper by S. Alexiou 2022, Analysis of pla sma emission experiments and “dips”

Several papers recently compared a spectroscopic model featuring dip structures to experimental spectra of hydrogenic ions obtained in the conditions of dense and hot plasmas created by lasers. The present paper by Alexiou carefully analyses the experimental conditions, the diagnostic using the dip model, and proposes new accurate calculations for the effect of periodic electric field on a line shape. This new model uses a Floquet eigendecomposition for the effect of the periodic field and/or static fields, and a computer simulation of the electron and ion microfields. Several different calculations are proposed, with and without fine structure, possibility of adding oscillating and static fields in a fixed direction. There is also a possibility to test the randomness in the turbulent electric field, and to investigate the effect of the dressing of the microfield potential energy. A useful feature is the calculation of the dipole autocorrelation functions for various components of the line.

Calculations presented concern the Lyman beta and gamma lines for the conditions mentioned in the experimental papers. Profiles of Lyman beta are calculated for 5 different plasma conditions and Lyman gamma for a single case. Each case is analyzed with a main calculation displaying the pi and sigma components for an oscillating field with a fixed direction and a non collinear static field. Other profiles investigate the effect of dressing by the oscillating field, the effect of the direction of the static field, the effect of randomness of a static field. Other calculations show the effect of cold ions and study the shape of the dipole autocorrelation functions C(t), excluding the occurrence of dips in the profile if C(t) stays positive.

All these calculated profiles are in the whole quite different from the experimental profiles. Multiple tests presented in the paper show that it seems impossible to reproduce the experimental profiles. Using the dips as was done in the experimental papers will thus not lead to a reliable diagnostic of the plasma and a possible turbulent field. The paper suggests a new experimental analysis including the effect of observation angle. It also suggests to introduce a sampling of the turbulent field in the analysis of experimental spectra. Including ion dynamics, and a dressing of all the fields is recommended.

The paper is well written, and presents clearly all the calculations performed and the discussions clearly question the initial experimental study.

I noticed a few typing errors

sattelite should be satellite

Line 41 Likewise?

Line 68 Blokhintsev (see also line 94, line 492)

Line 518 complicated

Author Response

I’d like to thank Referee 1 for his review. I have also implemented the changes requested:

Although Linewise is what I meant to say, I have rephrased it. I also corrected the numerous misspelings of Blokhintsev and complicated.

Reviewer 2 Report

The manuscript

“Analysis of plasma emission experiments and dips”

by S. Alexiou

reports about line shape calculations of highly charged ions in dense plasmas where turbulent fields result in a considerable modification of the spectral distribution within the line. The study of the turbulent field, and, in particular the characterization of the turbulence via emission line shapes are of great interest in laser-plasma interaction experiments.

The manuscript, however, is not well organized and need considerable revision:

            The discussion is rather opaque and contains a continuous mixture and bouncing between the author’s calculations of some restricted cases, critics to other works and numerous difficult to read intermediate mini-summaries/case studies.

            As the manuscript stands, it looks like a critical comment to the analysis presented in previous publications but it extends over more than 30 pages. This is inacceptable.

            The manuscript is also rather sloppy written and contains many typos and imprecisions. It is often not clear what the author is speaking about (e.g. it is not clear what it means “dip structures in plasmas”. What is certainly meant is a modification of the line profiles showing maxima and minima……).

The studied line profiles should be designated as Lyman-beta and not as L-beta…Lyman-gamma and not L-gamma etc..

Satellites: NOT sattelites

Recommendations:

1) Revise careful imprecisions and typos throughout the manuscript.

2) The manuscript should start with a clear introduction to the subject, namely line profiles in dense plasmas, Stark effect, line profile modifications of standard broadening theory due to turbulence, oscillating fields..etc.

In this respect, not only electric field but also magnetic fields should be mentioned.

3) The current reference list is rather poor and important papers of other authors on line shapes are not mentioned, in particular (but not exclusively)

- concerning dips in line profiles:

Gravilenko et al., Phys. Rev. A 73, 013202 (2006)

- concerning line profiles with oscillating field

Peyrusse, Phys. Scripta 56, 371 (1997)

- Lisitsa, Atoms in Plasmas, Springer 1994

4) After introduction, the present theory should follow and results of simulations should be presented while clear dependences on parameters should be worked out with figures that are self-explanatory (with clear legends and designation of curves rather then employing forth and back references to table values and case studies).

5) A discussion of the analysis of experimental data via perturbed line profiles (general analysis) should follow including the fact that usually not all parameters are known and often those which are known are not very precise.

5) A more careful and transparent analysis of the experiment and theory under criticism (namely those of refs. 1 and 2) should be provided.

E.g., concerning the results of the FLYCHK simulations. The reference 1 indicates ne=6x10**23 cm-3, kTe=500 eV. What line profiles would results from the present theory for these parameters ?

The experimental data from references 1 and 2 claim a very dense plasma, what is the impact of opacity on the line shapes ? Is differential plasma motion an issue ?

Lyman-beta could be also affected by so-called dielectronic satellites, what could be their influence on maxima and minima ?

What is also insufficiently discussed, is the precision of the experiment itself. The original data from ref. 1 and 2 look rather noisy and the author should discuss, whether these are significant variations that he analyses and criticizes or not.

What is likewise insufficiently discussed, is the spatial dependence of the spectra and the time evolution as the spectra presented in refs. 1 and 2 seem to be space and time integrated.

6) The authors should devote more effort to a transparent analysis of the theory presented in refs. 1 and 2 and contrast with his own theory/simulations rather than providing a continuous mixture of theory and experiment.

7) The author should also attempt to analyze Lyman-alpha and contrast his simulations with the high precision data presented in the work of Gravilenko et al 2006.

As the modifications to be made are rather numerous and profound, a second round of referee process is probably necessary after revision.

Author Response

First, I'd like to thank the reviewer for the comments. I have corrected the numerous spelling errors and hopefully clarified the scope of the present work. Indeed, there seems to have  been a major misunderstanding concerning the point of the present work. This is addressed now more clearly in the Introduction. Briefly, the point is not to match the experimental data (which can be done without invoking turbulence-indeed in my view there is no compelling evidence for suggesting turbulence-), nor can the present work address the valid points raised by the referee on opacity, noise, and spatial and temporal integration, though these are alluded to in the manuscript. Indeed, the present work cannot provide or clarify experimental details that the authors involved in the experiment did not provide. What the manuscript DOES show though is that the premise that these ‘dips’ are robust enough to overcome all these, noise, opacity, and spatial and temporal integration is not tenable.

Furthermore, the revised manuscript takes into account the referee’s comments, specifically:

-The introduction references previous work, including magnetic fields and clarifies the purpose of this work. Certainly I can give lots more references, but I do not consider them relevant to the point of the paper and do not wish to add ‘filler’ material. This would also make the manuscript longer.

- Typos are corrected( ‘Sloppy’ is unclear to me if it refers to anything other than ‘typos and imprecisions’.)

-’Dip structures’ in the abstract is clarified

-L-beta and L-gamma are standard notations , used among others, by the publications referenced in the same journal. So I do not see any need for this. Nevertheless, I have introduced this notation already in the interaction, which among other things helps save space.

Points where I do not agree with the referee are the following:

1)With regard to length:

As explained in the revised version, the manuscript’s purpose is both to correct these ‘previous publications’ and to show how the analysis should be done correctly. Also note that these ‘previous publications ‘ take more space without the level of detail in the present publication. It is very clear why the space taken is needed-just look at all the scenaria covered and their rationale. The referee does not explain what is ‘unacceptable’ about this. I am unaware of any rule that states that critical comments (which are typically short precisely because they do NOT usually involve elaborate and lengthy calculations) have to have a specified length or not to exceed it. The logical rule would be that any scientific work must take as much space as necessary to make its point, no more , no less. The manuscript certainly adheres to this rule of logic and the referee does not seem to dispute this.

I also note that adding an analysis of the Gavrilenko et. al. Paper, which is a very different experiment with different evidence for turbulent fields and would further add to the length-substantially I would add.

2) In my view (and in the view of proofreaders) since each case is a separate section, it is clearer to refer to ‘CaseA’, ‘Case B’, etc in the figures, as the important thing is the differences between the various calculations for each case. In my view this is simpler and much more convenient than if the reader sees the plasma and field parameters in each figure, then has to go back and check what the parameters were in the previous calculation. Again, the key is that the aim is not reproducing the experimental profile, but to show the dependencies on the parameters studied.

3)The reason I have not allocated more space to the ‘theory’ behind the ‘dip’ interpretation is that that theory is rather sketchy and skips a number of very important aspects. But mostly because doing so would take even more space to repeat the same explanations here. Thus, I am citing a few appropriate references. An analysis of the Gavrileko-Oks theory of 'dips' and its reduction to the present case will be unavoidably quite long and is best left to a different forthcoming paper, based on work presented in the recent Spectral Line Shapes Workshop, which will also analyse a number of general aspects of lineshapes in periodic fields. 

Reviewer 3 Report

The title is unclear; specifically, what is a "plasma emission experiment"?

In the abstract, "Recent experiments [...] have claimed" - experiments do not
claim; it is the authors of the respective papers who do. Similarly, the first
sentence in the main body and probably a few more places throughout the
manuscript. Further below, "Experiments do measure [...] the targets". Maybe the papers describe the target properties?

The overall structure of the manuscript is somewhat unusual. There is no
"Introduction", so the reader is left with no context.

I must admit that I do not trust the results presented in Refs. [1,2] - neither
the modeling, nor the experimental data themselves (i.e., the small features atop spectral lines that are supposed to be the special "dips" could easily be mere noise or experimental artifacts due to, e.g., defects in the X-ray spectrometer crystal). As Carl Sagan said, "extraordinary claims require extraordinary evidence", while the evidence presented is actually sub-standard. Some shortcomings are listed in the present manuscript.

Having said that, nevertheless, I find it unfair not to provide the details of
the model being criticized; there should probably be a reasonable chunk of text
devoted to its description.

Why do the simulations use a separate Floquet technique for the periodic fields
instead of doing the calculations with the total (plasma + periodic) fields? Is
there any gain? In any case, a comparison of the results obtained from the two
approaches would be interesting to see.

In many figures, there are "Experiment" spectra, but it is not clear which
exactly. Table 1 is probably the right place to make the correspondence. Unless
I am mistaken, Case A-D correspond to Shot A-D in Ref. [1], respectively, while
Case E - to Fig.1b of Ref. [2]b, correct?

Frankly, I see no point in discussing/providing results for so many very similar
cases. A-C (and E) are nearly the same from the point of view of density,
temperature, and fields. It is pretty obvious that if one of them fails to
reproduce the observed spectrum entirely, all others also do. This would
significantly reduce the volume of the very long paper without omitting anything
important/new.

The spectral range used for all "Ly-b" comparisons is too narrow, often cutting
the theoretical spectra. It seems this is the range of the experimental setup
from Ref. [1], but why is it preserved here??

Talking about the similarity of some cases, e.g., A and B, it is suspicious that
the calculated spectra in Figs. 2 and 11 differ so radically. What is the reason
for such a substantial "blue" shift? Again, the reader is left wondering what
has been cut to the left of the spectral range.

Sec. 3.1.5: 1-eV ion temperature means a very strong ion coupling. I doubt that
the plasma model used (independent Debye quasi-particles) remains valid.

The proper reference for FLYCHK is <https://doi.org/10.1016/j.hedp.2005.07.001>
(and should be cited after the first mention of the code); Ref. [4] is for FLY.

A number of spelling issues/typos were observed, e.g., "shown be sensitive" ->
"shown TO be sensitive", "frequency field amplitude" -> "frequency AND field
amplitude", "width if the dip" -> "width OF the dip", "Experments" ->
"Experiments", "Linewise" -> "Likewise", etc.

Author Response

I'd like to thank the reviewer for his efforts. Indeed, I have corrected the typos and/or expressions that might not be clear or accurate , such as  'plasma emission experiments' and 'experiments claim' and clarified the scope of the work, which was a major issue. I also added an introduction.  I also added the correspondence of each case to the figures of the references in Tables 1 and 2.

Of course I share the view the reviewer as well as of a number of colleagues(“I do not believe in analyzing noise” was one colleague’s view) on the experiments. Nevertheless, these results and ‘interpretation’ have been published in a number of places and because it is advertised as ‘simple’, it is likely to mislead more researchers in the future. So the point of this work is twofold: Set the record straight by showing that these 'dips' are NOT some sort of robust signatures that are independent of directionality, variations, temporal and spatial integration etc  and show how this analysis should be done correctly.

The reason the manuscript does not go into more detail of the ‘model’ being critisized is twofold: First, there is not much more detail for the model, as presented by its proponents-details are sketchy at best. There are hardly any explanations. Second, expaining all the assumptions and approximations and their appropriateness of the Gavrileko-Oks dips theory will be unavoidably quite lengthy on a paper that is already long (see below on the referee's suggestion that some calculations might be similar).   This will be part of a forthcoming paper based on the recent Spectral Line Shapes Workshop and will include general aspects of lineshapes with oscillatory fields, of which this analysis will be one.  

The referee also asks why use  the  interaction picture. I have explained this in Ref.5 of the original submission and 16 in the revised version(identify the new features and avoid stiffness-namely use a stiff solver to solve the periodic field ONCE and a non-stiff solver to solve for each  random plasma configuration). In addition, this issue was  addressed at the very recent (October 22) Spectral Line Shapes Conference and will be published separately along with general aspects. Briefly, normal line shape calculations compute dab.db’a’{Uaa’(t)Ubb’+(t)}with {…} denoting average. So if one has a code that solves the U-matrices in the presence of the plasma fields, why not just add the external static and oscillatory fields? The problem is that this form assumes stationary processes and is inapplicable here. This can be easily verified by neglecting the plasma fields and assuming H-like emitters just a linearly polarized field. The standard approach that assumes stationary processes results in a profile of delta functions with intensities J_p
, whereas the correct result-obtained by the method in the calculations presented which does NOT assume stationary processes- is the known Blokhintsev result , which involves squares of Bessel functions J_p, and does NOT give negative profiles, while the form with J_p can.

A brief list of referee suggestions I did not implement:

-I do not agree or understand why the referee expected the various Lyman-beta cases to be 'similar'.  Same for the 'blue shift': The solution of the 0th order Hamiltonian(i.e. without the plasma electrons and ions) is certainly non-perturbative and  the peak positions vary significantly. I have added a note and a graph to this effect. Indeed the blue shift noted by itself explains why all cases had to be computed.

-Regarding the spectra range covered,  any calculation must in one way or another define a spectral range of interest. This was defined as the range in the experiment and/or the calculations accompanying the experiments. What would be the point in extending that range when neither the experiment nor previous calculations show the extra region suggested?

-"Maybe the papers describe the target properties?"

I think as suggested, it would be less clear. What properties? Before or after irradiation? Because they do not describe the properties before and the properties after depend on the irradiation, hence they are not property of the target.

Round 2

Reviewer 2 Report

-